# Proteotoxicity from aberrant ribosome biogenesis compromises cell fitness

Blake W Tye[1,2], Nicoletta Commins[3], Lillia V Ryazanova[4,5], Martin Wühr[4,5], Michael Springer[3], David Pincus[6,7,8], L Stirling Churchman[1]*

[1]Department of Genetics, Harvard Medical School, Boston, United States; [2]Program in Chemical Biology, Harvard University, Cambridge, United States; [3]Department of Systems Biology, Harvard Medical School, Boston, United States; [4]Department of Molecular Biology, Princeton University, Princeton, United States; [5]Lewis-Sigler Institute for Integrative Genomics, Princeton University, Princeton, United States; [6]Whitehead Institute for Biomedical Research, Cambridge, United States; [7]Department of Molecular Genetics and Cell Biology, University of Chicago, Chicago, United States; [8]Center for Physics of Evolving Systems, University of Chicago, Chicago, United States

**Abstract** To achieve maximal growth, cells must manage a massive economy of ribosomal proteins (r-proteins) and RNAs (rRNAs) to produce thousands of ribosomes every minute. Although ribosomes are essential in all cells, natural disruptions to ribosome biogenesis lead to heterogeneous phenotypes. Here, we model these perturbations in *Saccharomyces cerevisiae* and show that challenges to ribosome biogenesis result in acute loss of proteostasis. Imbalances in the synthesis of r-proteins and rRNAs lead to the rapid aggregation of newly synthesized orphan r-proteins and compromise essential cellular processes, which cells alleviate by activating proteostasis genes. Exogenously bolstering the proteostasis network increases cellular fitness in the face of challenges to ribosome assembly, demonstrating the direct contribution of orphan r-proteins to cellular phenotypes. We propose that ribosome assembly is a key vulnerability of proteostasis maintenance in proliferating cells that may be compromised by diverse genetic, environmental, and xenobiotic perturbations that generate orphan r-proteins.
DOI: https://doi.org/10.7554/eLife.43002.001

*For correspondence:
churchman@genetics.med.
harvard.edu

Competing interests: The authors declare that no competing interests exist.

## Introduction

Ribosomes are large macromolecular machines that carry out cellular protein synthesis. Cells dedicate up to half of all protein and RNA synthesis to the production of ribosomal protein (r-protein) and RNA (rRNA) components required to assemble thousands of new ribosomes every minute (*Warner, 1999*). rRNAs and r-proteins are coordinately synthesized and matured in the nucleolus and cytosol, respectively, in response to growth cues (*Lempiäinen and Shore, 2009*). R-proteins are co- and post-translationally folded, requiring general chaperones as well as dedicated chaperones called escortins (*Pillet et al., 2017*). Thus, ribosome assembly requires the coordinated synthesis and assembly of macromolecules across cellular compartments, and must be performed at extremely high rates.

The balanced synthesis of rRNA and r-protein components in proliferating cells is frequently disrupted by genetic and extracellular insults, leading to a wide range of phenotypes. Environmental stressors, such as heat shock and viral infection, and xenobiotics, such as DNA-damaging agents used as chemotherapeutics, interfere with rRNA processing and nucleolar morphology (*Burger et al., 2010*; *Kos-Braun et al., 2017*; *Liu et al., 1996*; *Pelham, 1984*). In zebrafish, and

**eLife digest** Cells are made up of thousands of different proteins that perform unique roles required for life. To create all of these proteins, cells use machines called ribosomes that are partly formed of elements known as r-proteins. When cells grow and divide, the ribosomes have to make copies of themselves through a process called ribosome biogenesis.

Although all cells need ribosomes, certain types of cells are especially sensitive to events that interfere with ribosome biogenesis. For example, patients that have mutations in genes needed for ribosome biogenesis produce fewer red blood cells, but their other cells and tissues are mostly healthy. It is not clear why some cells are more sensitive than others.

Ribosome biogenesis is very similar between different organisms, so researchers often use budding yeast as a model to study the process. Here, Tye et al. used genetic and chemical tools to interfere with ribosome biogenesis on short time scales, which made it possible to detect early on what was going wrong in the cells.

The experiments found that when ribosome biogenesis was disrupted, r-proteins that were waiting to be assembled into ribosomes quickly stuck to one another and formed clumps that reduced the ability of the yeast cells to grow. The cells responded by switching on a protein called Hsf1, which restored their ability to grow. Yeast cells that were growing quickly, and therefore making more ribosomes, were more sensitive to abnormal ribosome biogenesis than slow-growing cells.

These results indicate that how actively a cell is growing, and its ability to cope with r-proteins sticking together, may in part explain why certain cells are more vulnerable to events that interfere with ribosome biogenesis. Since human cells also have Hsf1, future experiments could investigate whether turning it on might also protect fast-growing human cells from such events.

DOI: https://doi.org/10.7554/eLife.43002.002

possibly in humans, hemizygous loss of r-protein genes can drive cancer formation (*Amsterdam et al., 2004*; *Goudarzi and Lindström, 2016*). Diverse loss-of-function mutations in genes encoding r-proteins, r-protein assembly factors, and rRNA synthesis machinery result in tissue-specific pathologies in humans (ribosomopathies), such as red blood cell differentiation defects in patients with Diamond–Blackfan anemia (DBA) (*Draptchinskaia et al., 1999*; *Khajuria et al., 2018*; *Narla and Ebert, 2010*). Not all the phenotypes caused by defects in ribosome biogenesis are wholly deleterious: in budding yeast, loss of r-protein genes increases stress resistance and replicative lifespan and reduces cell size and growth (*Jorgensen et al., 2004*; *Steffen et al., 2008*, *Steffen et al., 2012*), and mutations in r-protein genes in *C. elegans* also extend lifespan. Collectively, then, despite the fact that ribosomes are required in all cells, disruptions in ribosome biogenesis lead to an array of phenotypic consequences that depend strongly on the cellular context.

Phenotypes resulting from perturbations to ribosome assembly have both translation-dependent and -independent origins. As expected, when ribosomes are less abundant, biomass accumulation slows and growth rates decreases. Furthermore, reduced ribosome concentrations alter global translation efficiencies, impacting the proteome in cell state–specific ways (*Khajuria et al., 2018*; *Mills and Green, 2017*). In many cases, however, cellular growth is affected before ribosome pools have appreciably diminished, indicating that perturbations of ribosome assembly have translation-independent or extraribosomal effects. The origins of these effects are not well understood, but may involve unassembled r-proteins. In many ribosomopathies, excess r-proteins directly interact with and activate p53, presumably as a consequence of imbalanced r-protein stoichiometry. However, p53 activation is not sufficient to explain the extraribosomal phenotypes observed in ribosomopathies or in model organisms experiencing disrupted ribosome biogenesis (*James et al., 2014*). Interestingly, r-proteins produced in excess of one-another are normally surveyed by a ubiquitin-proteasome-dependent degradation (*McShane et al., 2016*), which appears to prevent their aberrant aggregation (*Sung et al., 2016a*; *Sung et al., 2016b*).

To determine how cells respond and adapt to perturbations in ribosome assembly, we took advantage of fast-acting chemical-genetic tools in *Saccharomyces cerevisiae* to rapidly and specifically disrupt various stages of ribosome assembly. These approaches capture the kinetics of cellular

responses, avoid secondary effects, and are far more specific than available fast-acting chemicals that disrupt ribosome assembly, such as transcription inhibitors, topoisomerase inhibitors, and nucleotide analogs. Furthermore, by performing this analysis in yeast, which lacks p53, we obtained insight into the fundamental, p53-independent consequences of perturbations of ribosome biogenesis.

We found that in the wake of perturbed ribosome assembly, cells experience a rapid collapse of protein folding homeostasis that independently impacts cell growth. This proteotoxicity is due to accumulation of excess newly synthesized r-proteins, which are found in insoluble aggregates. Under these conditions, cells launch an adaptive proteostasis response, consisting of Heat Shock Factor 1 (Hsf1)-dependent upregulation of chaperone and degradation machinery, which is required for adapting to r-protein assembly stress. Bolstering the proteostasis network by exogenously activating the Hsf1 regulon increases cellular fitness when ribosome assembly is perturbed. The high degree of conservation of Hsf1, proteostasis networks, and ribosome assembly indicates that the many conditions that disrupt ribosome assembly and orphan r-proteins in other systems may also drive proteostasis collapse, representing a key extraribosomal vulnerability in cells with high rates of ribosome production.

## Results

### Imbalanced rRNA:r-protein synthesis elicits upregulation of proteostasis machinery via heat-shock factor 1 (Hsf1)

Ribosome biogenesis commences in the nucleolus, where rRNA is synthesized and processed, and many r-proteins are assembled concomitantly (*Figure 1A*). As a first class of disruption to ribosome biogenesis, we examined the consequences of imbalances in rRNA and r-protein production. Specifically, we focused on nuclease factors involved in several different stages of processing rRNAs for the large (60S) ribosomal subunit: endonuclease Las1, 5'-exonucleases Rat1 and Rrp17, and 3'-exonuclease Rrp44/Dis3 (exosome) (*Kressler et al., 2017*; *Turowski and Tollervey, 2015*; *Woolford and Baserga, 2013*). We tagged the target molecules with an auxin-inducible degron (AID), which allows rapid depletion of a tagged protein upon addition of the small molecule auxin (*Nishimura et al., 2009*), thereby acutely shutting down production of mature rRNA (*Figure 1B*). The rRNA processing factors were depleted by 75–90% within 10–20 min of auxin addition, and precursor rRNA (pre-rRNA) accumulated by 20 min, confirming that depletion of these factors rapidly interfered with rRNA processing (*Figure 1C,D*). Depletion also led to a detectable reduction in the level of free 60S subunits, indicating that the cell was failing to assemble new 60S, but had no effect on the mature ribosome pool (*Figure 1—figure supplement 1A*).

To determine whether cells respond directly to disrupted rRNA production, we explored the immediate transcriptional response following depletion of these factors. For this purpose, we auxin-treated (or mock-treated) each strain for 20 min, and then performed gene expression profiling by RNA-seq. WT cells exhibited no alteration of the transcriptome in the presence of auxin, whereas each AID-tagged strain exhibited the same compact response. Remarkably, the induced genes are known targets of Heat-Shock Factor 1 (Hsf1), a conserved master transcription factor that controls protein folding and degradation capacity in stress, aging, and disease (*Akerfelt et al., 2010*) (*Figure 1E*). Hsf1 directly controls ~50 genes encoding proteostasis factors, including protein folding chaperones (*SSA1/4* (Hsp70), *HSP82* (Hsp90), co-chaperones), aggregate clearance factors (*BTN2*, *HSP42*, *HSP104*), the transcription factor that regulates proteasome abundance (*RPN4*), and ubiquitin (*UBI4*) (*Pincus et al., 2018*; *Solís et al., 2016*). Upregulation of Hsf1-dependent genes coincided with an increase in Hsf1 occupancy at their promoters (*Figure 1—figure supplement 1B*) and was independent of the translational stalling pathway (Rqc2, *Figure 1—figure supplement 1C*). Hsf1-target transcripts, measured by Northern blot, were maintained at high levels over an 80 min time-course of auxin treatment (*Figure 1—figure supplement 1D*). AID-tagged Rrp17 acted as a partial loss-of-function allele, as indicated by the accumulation of pre-rRNA even in the absence of auxin and reduced cell growth (*Figure 1D* and data not shown), potentially explaining the mild and more transient upregulation of Hsf1 target transcripts following auxin addition in the strain expressing this protein. Nevertheless, depletion of all four rRNA processing factors each led to strong and specific activation of the Hsf1 regulon.

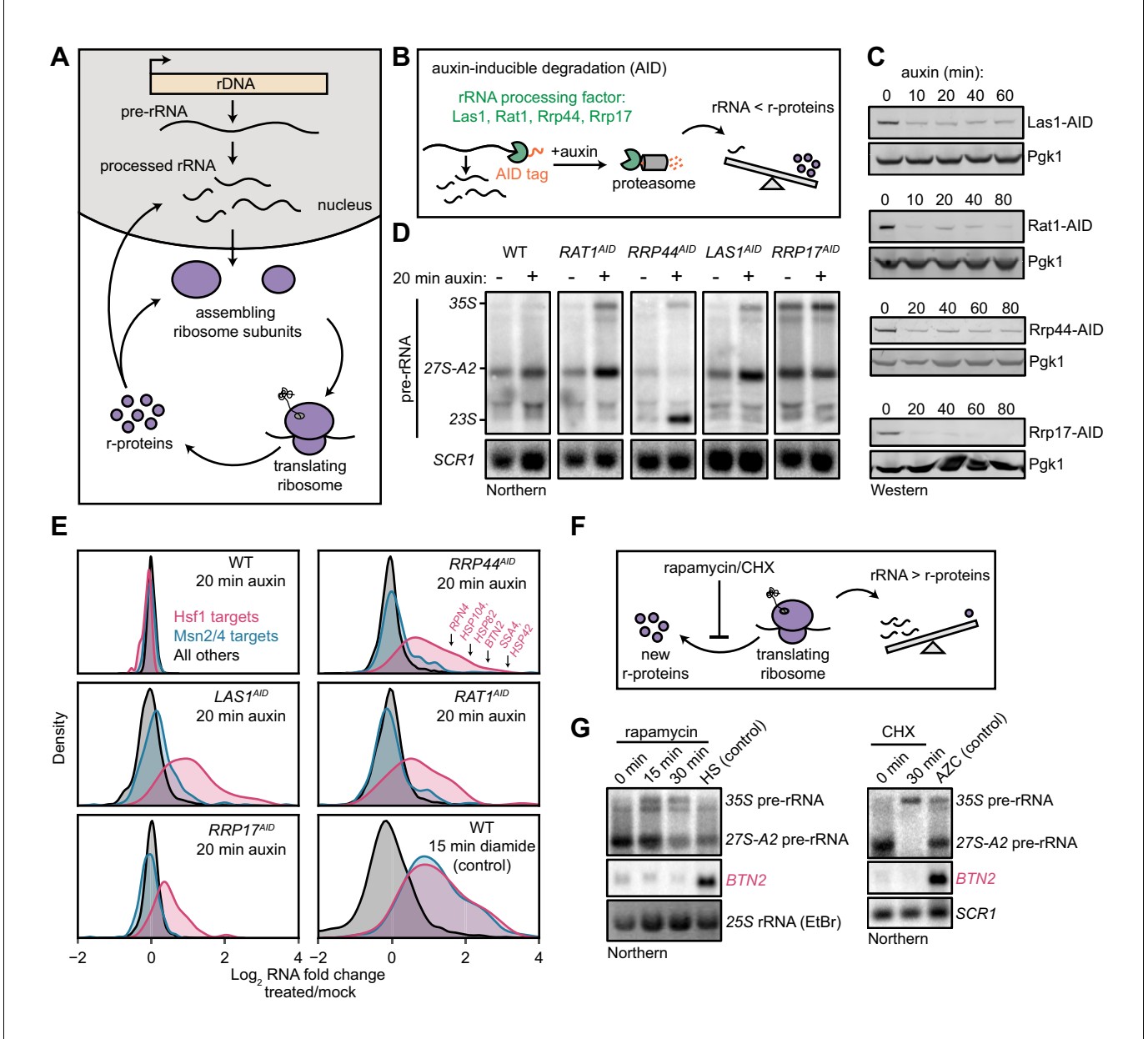

**Figure 1.** Imbalanced rRNA:r-protein synthesis elicits upregulation of proteostasis machinery via Heat-Shock Factor 1 (Hsf1). (**A**) Brief schematic overview of ribosome biogenesis. (**B**) Auxin-inducible degradation (AID) of rRNA processing factors. The C-terminus of the protein is genetically tagged with the AID tag (IAA7-V5) in cells co-expressing the E3 ligase adapter *Os*TIR1. Addition of auxin allows recognition and degradation of AID-tagged proteins by the proteasome. (**C**) Depletion of AID-tagged rRNA processing factors following addition of auxin (100 μM) detected by anti-V5 immunoblot. (**D**) Pre-rRNA accumulation following rRNA processing factor depletions. RNA from mock and auxin (20 min) treated cells was analyzed by Northern blot with a probe (800, see ***Supplementary file 3***) that recognizes full-length pre-rRNA (35S) and processing intermediates (27S-A2 and 23S) (***Kos-Braun et al., 2017***). (**E**) Upregulation of Hsf1 targets in rRNA processing factor-depleted cells. RNA-seq density plots of $\log_2$ fold change after 20 min auxin treatment (versus mock-treated control), determined from n = 2 biological replicates. Hsf1 targets, n = 42; Msn2/4 targets, n = 207; all others, n = 4912. The oxidative agent diamide (15 min, 1.5 mM) was used as a comparative control. The WT strain treated with auxin also expressed OsTIR1 but lacked any AID-tagged factor. (**F**) Schematic illustrating that rapamycin and CHX treatment acutely shutdown r-protein synthesis ahead of rRNA synthesis leading to an imbalance in ribosome components. (**G**) Northern blots of pre-rRNA and Hsf1-dependent *BTN2* from WT cells treated with rapamycin (200 ng/ml) or cycloheximide (CHX, 200 μg/ml) for the indicated times. Heat shock (HS, 37°C, 15 min) and azetidine-2-carboxylic acid (AZC, 10 mM, 30 min) were used as positive controls for Hsf1 activation.

DOI: https://doi.org/10.7554/eLife.43002.003

The following figure supplements are available for figure 1:

**Figure supplement 1.** Kinetics of Hsf1 activation.

*Figure 1 continued on next page*

*Figure 1 continued*

DOI: https://doi.org/10.7554/eLife.43002.004

**Figure supplement 2.** Specificity of Hsf1 activation by depletion of rRNA processing factors.

DOI: https://doi.org/10.7554/eLife.43002.005

Importantly, we ruled out the possibility that the depletion strategy itself resulted in Hsf1 activation. Depletion of several factors not involved in rRNA processing via AID did not activate Hsf1, including the RNA surveillance exonuclease Xrn1, mRNA decapping enzyme Dxo1, and transcription termination factor Rtt103 (*Figure 1—figure supplement 2A,B*). Additionally, nuclear depletion of an rRNA processing factor using an orthogonal method that does not require proteasome-mediated degradation ('anchor-away') (*Haruki et al., 2008*) likewise led to Hsf1 activation, whereas anchor-away depletion of another nuclear protein did not (*Figure 1—figure supplement 2C–F*).

Stress conditions and xenobiotics in yeast characteristically activate a 'general' environmental stress response (ESR), driven by the transcription factors Msn2/4, which rewires metabolism and fortifies cells against further stress (*Gasch et al., 2000*). Strikingly, Msn2/4-dependent ESR genes were not activated after depletion of rRNA processing factors (*Figure 1E*). By contrast, treatment of WT cells with the oxidative agent diamide for 15 min potently activated both Hsf1- and Msn2/4-dependent genes, as expected (*Figure 1E*). Highly specific activation of Hsf1 in the absence of ESR has only been observed in circumstances in which cellular proteostasis is acutely strained: treatment with azetidine-2-carboxylic acid (AZC), a proline analog that interferes with nascent protein folding, resulting in aggregation (*Trotter et al., 2002*), or overexpression of an aggregation-prone mutant protein (*Geiler-Samerotte et al., 2011*). Comparison of the kinetics of pre-rRNA and Hsf1-dependent transcript accumulation revealed that cells activate Hsf1 within minutes after rRNA processing is disrupted, indicating a rapid strain on proteostasis, as observed in instantaneous heat shock (*Figure 1—figure supplement 1E*).

The results of acute disruption of rRNA processing suggest that Hsf1 is activated by an excess of newly synthesized r-proteins relative to rRNAs. To determine whether the reverse phenomenon (i.e. a surplus of rRNAs relative to new r-proteins) could also activate Hsf1, we treated cells with rapamycin to inhibit r-protein expression by inactivating TORC1 (*Figure 1F*). During the first 15–30 min of low-dose rapamycin treatment, cells strongly repress synthesis of r-proteins while maintaining normal levels of rRNA transcription (*Reiter et al., 2011*). Precursor rRNA accumulated due to r-protein limitation, as expected, but the Hsf1-dependent gene *BTN2* was not upregulated during rapamycin treatment (*Figure 1G*). Similarly, halting translation, and thus r-protein synthesis, with cycloheximide (CHX) resulted in pre-rRNA accumulation but no upregulation *of BTN2*. On the basis of these findings, we conclude that when r-proteins are in excess relative to what can be assembled into ribosomes, yielding orphan r-proteins, cells activate a proteostatic stress response driven by Hsf1.

## Orphan r-proteins are sufficient to activate the Hsf1 regulon

As an orthogonal means of testing the model that orphan r-proteins activate the Hsf1 regulon, we directly inhibited assembly of r-proteins. To this end, we treated cells with a small molecule, diazaborine (DZA), that blocks cytoplasmic assembly of several r-proteins into the 60S subunit by specifically inhibiting the ATPase Drg1 (*Loibl et al., 2014*) (*Figure 2A*). Screens for DZA resistance have yielded only mutations in factors involved in drug efflux and the gene encoding the drug's mechanistic target, *DRG1*, indicating that the compound is highly specific (*Wendler et al., 1997*). Over a time-course of moderate, sublethal DZA treatment, the Hsf1-dependent transcripts *BTN2* and *HSP82* strongly accumulated by 15 min, whereas the Msn2/4-dependent transcript *HSP12* exhibited no response (*Figure 2B*). Moreover, Hsf1-dependent transcripts returned to basal levels at 90 min, indicating that Hsf1 activation was an adaptive response. Importantly, a DZA-resistant point mutant of Drg1 (V725E) (*Loibl et al., 2014*) restored cell growth and reduced accumulation of Hsf1-dependent transcripts, confirming that DZA contributes to Hsf1 activation via the expected mechanism (*Figure 2—figure supplement 1*). Consistent with a functional role of Hsf1 activation, we found that DZA treatment protected cells from subsequent lethal heat stress (thermotolerance) (*Figure 2—figure supplement 2*). In cells treated with DZA for 15 or 45 min, RNA-seq revealed activation of the same response that was induced by depletion of rRNA processing factors: upregulation of Hsf1-

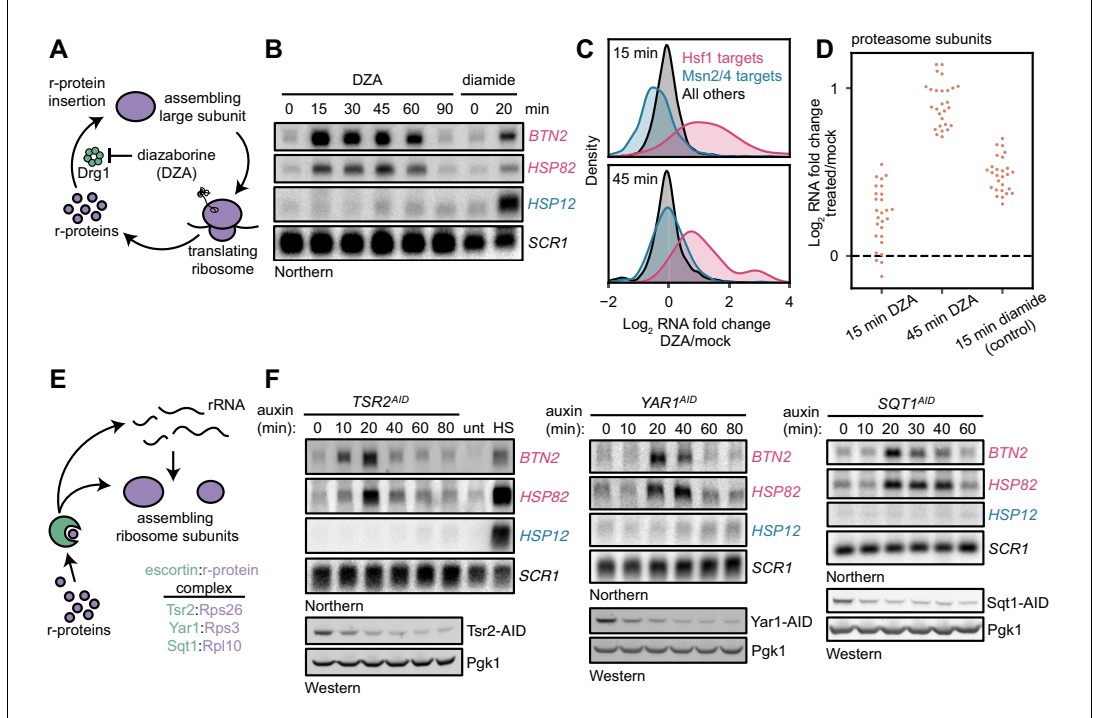

**Figure 2.** Orphan r-proteins are sufficient to activate the Hsf1 regulon. (**A**) Schematic describing that diazaborine (DZA) inhibits Drg1, preventing r-protein assembly into pre-60S subunits. (**B**) Kinetics of Hsf1 activation following DZA treatment. Northern blot of Hsf1-dependent *BTN2* and *HSP82* and Msn2/4-dependent *HSP12* transcripts from cells treated with DZA (15 μg/ml) for the indicated time. Diamide (1.5 mM) was used as a positive control for Hsf1 and Msn2/4 activation. (**C**) Upregulation of Hsf1 targets in DZA-treated cells. RNA-seq density plots of log$_2$ fold change after 15 or 45 min DZA treatment (versus DMSO-treated control), determined from n = 2 biological replicates. (**D**) Upregulation of proteasome subunits during RPAS. Swarm plot of log$_2$ fold change after 15 or 45 min DZA or 15 min diamide treatment for transcripts encoding proteasome subunits (n = 27). (**E**) Schematic describing how escortins Tsr2, Yar1, and Sqt1 chaperone newly synthesized Rps26, Rps3, and Rpl10, respectively, to assembling ribosomes. (**F**) Western blots showing depletion of AID-tagged Tsr2, Yar1, and Sqt1 and Northern blots for Hsf1-dependent *BTN2* and *HSP82* and Msn2/4-dependent *HSP12* transcripts at the indicated time after auxin addition. Unt, untreated; HS, heat shock.

DOI: https://doi.org/10.7554/eLife.43002.006

The following figure supplements are available for figure 2:

**Figure supplement 1.** On-target inhibition of Drg1 by DZA.
DOI: https://doi.org/10.7554/eLife.43002.007
**Figure supplement 2.** DZA treatment enhances thermotolerance.
DOI: https://doi.org/10.7554/eLife.43002.008
**Figure supplement 3.** The endoplasmic reticulum unfolded protein response (UPR) is not activated during RPAS.
DOI: https://doi.org/10.7554/eLife.43002.009

dependent proteostasis genes in the absence of Msn2/4-dependent general stress genes (*Figure 2C*). Furthermore, by 45 min, cells upregulated proteasome subunits ~ 2 fold, consistent with the early Hsf1-dependent upregulation of the proteasome-regulatory transcription factor *RPN4* (*Figure 2D*) (*Fleming et al., 2002*). Consistent with the exceptional specificity of this perturbation in eliciting an Hsf1-dependent response, we found that the canonical unfolded protein response (UPR), which responds to misfolded proteins in the endoplasmic reticulum, was not activated by either DZA or depletion of rRNA processing factors (*Figure 2—figure supplement 3*).

As another means to inhibit r-protein assembly, we depleted dedicated r-protein chaperones, called escortins (*Kressler et al., 2012*; *Pillet et al., 2017*). Each escortin binds a specific newly synthesized r-protein and brings it to the assembling ribosome, preventing aberrant aggregation (*Figure 2E*). We generated AID-tagged strains for the Rps26 escortin Tsr2, whose mutation in human cells leads to DBA (*Khajuria et al., 2018*). We also analyzed two other escortins, Sqt1 (Rpl10) and Yar1 (Rps3), and performed a time-course of auxin treatment for all three. Each escortin was depleted ~70% by 20 min. Northern blots revealed accumulation of *BTN2* and *HSP82* mRNAs by

10–20 min, with no change in the level of Msn2/4-regulated *HSP12* mRNA (*Figure 2F*). Both Rps26 and Rps3 are assembled into the pre-40S in the nucleus, whereas Rpl10 is the last r-protein assembled into the ribosome in the cytoplasm. Thus, either by inhibition of Drg1 or depletion of escortins, orphan r-proteins are sufficient to activate the Hsf1 regulon. Accordingly, we refer to the stress imparted by orphan r-proteins as ribosomal protein assembly stress (RPAS).

## Compromised r-protein gene expression and translational output during RPAS

In addition to the upregulation of the Hsf1 regulon in RPAS, we also observed downregulation of some genes. Intriguingly, the set of downregulated genes comprised mostly r-protein genes (*Figure 3A,B*). Under many stress conditions, both r-protein genes and assembly factor genes, collectively termed the ribosome biogenesis (RiBi) regulon, are repressed through Tor-dependent signaling (*Jorgensen et al., 2004*; *Marion et al., 2004*; *Urban et al., 2007*) (e.g. oxidative stress by diamide, *Figure 3A,B*). Therefore, we suspected that the specific downregulation of r-protein genes, but not assembly factors, in RPAS would not be executed through Tor. Indeed, cells treated with DZA for 15 or 45 min exhibited no change in the level of the TORC1 activity reporter, phosphorylated (phos-) Rps6 (*González et al., 2015*) (*Figure 3F*).

Many stress conditions lead to global translational repression, mediated in part by the kinase Gcn2, and enable specialized or cap-independent translation programs that aid in coping with the stress (*Wek, 2018*). Previous experiments with DZA showed that translation is downregulated shortly after treatment (*Pertschy et al., 2004*). To determine whether translation is repressed in RPAS, we monitored the synthesis of various V5-tagged ORFs. Transcription of V5-tagged transgenes was activated by the synthetic transcription factor Gal4–estradiol receptor (ER)–Msn2 activation domain (AD) (GEM) upon the addition of estradiol (*Stewart-Ornstein et al., 2012*) (*Figure 3C*). Under normal conditions, we found that the V5-tagged proteins began to accumulate after 10 min (*Figure 3D*). To determine the effect of RPAS on translational output, we briefly treated ORF-V5 strains with estradiol followed by DZA for 20 min and assessed the level of protein accumulation. All ORFs, including GFP-V5, accumulated to lower levels when cells were treated with DZA, consistent with a rapid reduction in translational output under RPAS (*Figure 3E*). Because DZA could achieve a maximal reduction of 20% in the ribosome pool in a 20-min experiment, this >50% reduction in synthesis cannot be explained by a diminishing ribosome pool. Interestingly, the reduction in translational capacity is not mediated through the kinase Gcn2 as in other stresses such as carbon or nitrogen starvation and oxidative stress, as phosphorylated (phos-) eIF2α did not accumulate during DZA treatment (*Cherkasova and Hinnebusch, 2003*; *Dever et al., 1992*; *Shenton et al., 2006*) (*Figure 3F*). In sum, we observed compromised r-protein gene transcription and global translational output during RPAS independent of canonical signaling pathways.

## Aggregation of orphan r-proteins during RPAS

Hsf1 responds to an increased prevalence of misfolded or aggregated proteins, and activates a transcriptional program to resolve these issues. Several r-proteins are found to aggregate in the absence of general cotranslational folding machinery, post-translational escortins, or nuclear import machinery (*Jäkel et al., 2002*; *Koplin et al., 2010*; *Pillet et al., 2017*). Further, excess r-proteins are targeted for degradation by Excess Ribosomal Protein Quality Control (ERISQ), a ubiquitin-proteasome-mediated pathway, in the absence of which r-proteins likewise prevalently aggregate (*Sung et al., 2016a*; *Sung et al., 2016b*). We therefore hypothesized that following disruptions to ribosome assembly, newly synthesized orphan r-proteins would aggregate. Supporting this idea, we found that Hsf1 activation by DZA required ongoing translation: pre-treatment with CHX prevented upregulation of Hsf1 targets, supporting the model of proteotoxic orphan r-proteins (*Figure 4A*). Similarly, Hsf1 activation by depletion of the rRNA processing factor Rat1 was fully inhibited by CHX pre-treatment (*Figure 4—figure supplement 1A*).

To test for the presence of protein aggregation in DZA-treated cells, we used a sedimentation assay that separates soluble proteins from large, insoluble assemblies (*Figure 4B*) (*Wallace et al., 2015*). As a positive control, we induced global protein misfolding by AZC and observed gross protein aggregates associated with disaggregases Hsp70 and Hsp104 (*Figure 4C*). By contrast, RPAS induced by DZA treatment resulted in no such gross protein aggregation, even at 40 min.

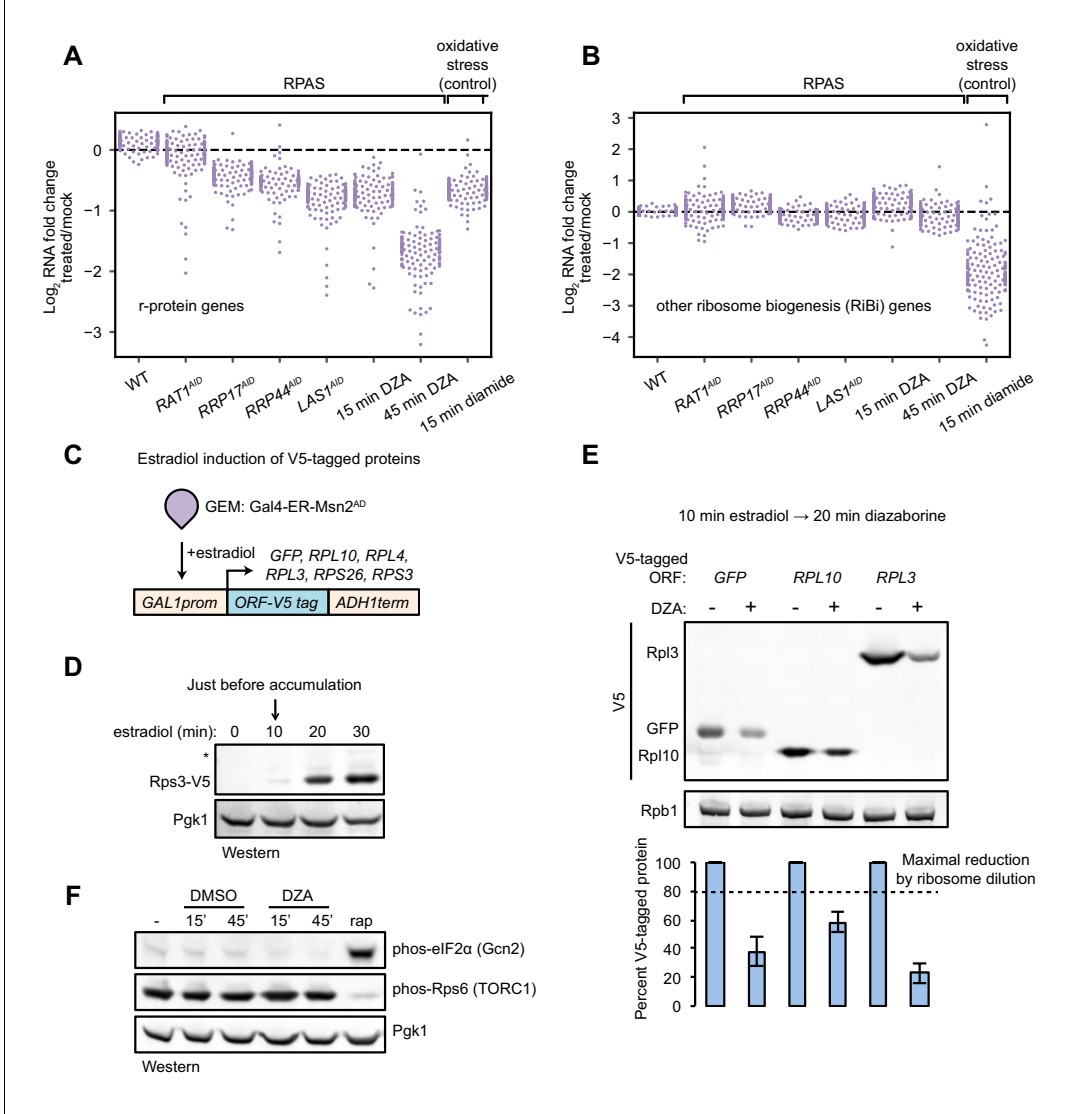

**Figure 3.** Compromised r-protein gene expression and translational output during RPAS. (A) Swarm plot of $\log_2$ fold change of r-protein encoding transcripts in the condition indicated on the x-axis (n = 136). (B) Swarm plot of $\log_2$ fold change of transcripts encoding ribosome biogenesis (RiBi) factors, excluding r-protein genes, in the condition indicated on the x-axis (n = 169). (C) Schematic of transgene system for estradiol-inducible expression of V5-tagged ORFs. (D) Western blot showing time-course of induction of Rps3-V5 after the indicated time of beta-estradiol (100 nM) addition. (E) Strains containing the indicated V5-tagged transgene were induced for 10 min with estradiol and then treated with vehicle (-) or 15 µg/ml DZA (+) for 20 min and analyzed by western blot (upper) and quantified relative to vehicle control (lower). Bar height indicates the average and error bars the standard deviation of n = 3 biological replicates. The dashed line corresponds to the hypothetical maximal reduction amount (to 80% of control) in protein produced as a result of ribosome dilution alone in 20 min (one fourth of a cell cycle). (F) WT cells were treated with vehicle (DMSO) or DZA for 15 or 45 min and analyzed by western blot. Rapamycin (rap, 200 ng/ml, 45 min) was used as a positive control for altering Gcn2 and TORC1 activity (*Dever et al., 1992*; *González et al., 2015*).

DOI: https://doi.org/10.7554/eLife.43002.011

We next asked whether newly synthesized r-proteins aggregated during RPAS. Using the estradiol induction system for V5-tagged ORFs, we followed the fate of newly synthesized r-proteins in mock- or DZA-treated cells. We found that newly synthesized Rps26, Rpl10, and Rpl3 shifted three- to fivefold to the insoluble fraction upon DZA treatment (*Figure 4D,F*). Interestingly, the levels of Rpl4 and Rps3 in the pellet increased modestly if at all, possibly due to their distinct biochemical characteristics, protection from aggregation by chaperones, or rapid assembly into precursor ribosome subunits. Treating extracts with the nuclease benzonase did not solubilize aggregated

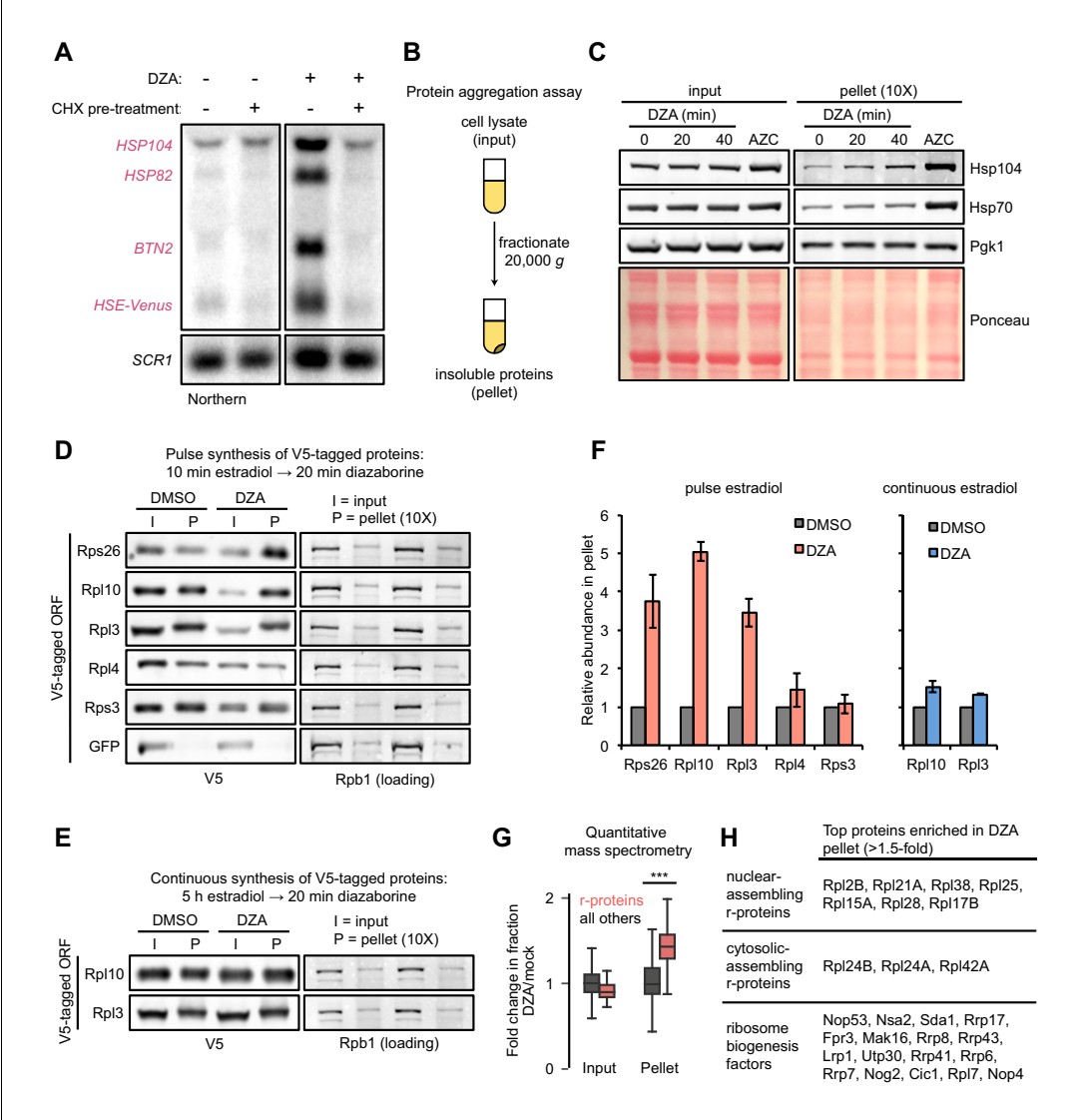

**Figure 4.** Aggregation of orphan r-proteins during RPAS. (**A**) Cells were mock or CHX (200 μg/ml) treated for 3 min prior to addition of DZA for 20 min and Hsf1 target were detected by Northern blot. *HSE-Venus*, *Venus* transgene downstream of four Hsf1-binding sites (Heat Shock Element, HSE). (**B**) Schematic of the protein aggregation assay. Proteins extracted from cryogenically lysed cells were fractionated by centrifugation at 20,000 *g* for 20 min to pellet insoluble proteins. (**C**) Cells were treated with DZA for 0, 20, or 40 min. Input and insoluble proteins (pellet) were resolved by SDS-PAGE. AZC (10 mM, 40 min) was used as a control to compare DZA results to a general increase in aggregates in the pellet, by Ponceau staining, and Hsp70 and Hsp104 sedimentation. 10X more of the pellet sample than input sample was loaded to increase sensitivity. (**D**) Strains expressing the indicated V5-tagged r-protein (or GFP as a control) were induced for 10 min with estradiol followed by vehicle (DMSO) or DZA treatment for 20 min. Input and pellet samples for all were analyzed by Western blot. 10X more of the pellet sample than input sample was loaded to increase sensitivity. (**E**) Same as (**D**), except cells were continuously induced for 5 hr with estradiol to label the mature protein pool prior to DMSO or DZA treatment. (**F**) Quantification of the indicated V5-tagged proteins in the pellet fraction versus the input (from panels D and E), normalized to the pellet to input ratio of Rpb1. The ratio was set to one for DMSO-treated cells. Bar height indicates the average and error bars the range of n = 2 biological replicates. (**G**) Box plot depicting results of quantitative mass spectrometry on proteins that pellet following 20 min mock (DMSO) or DZA treatment. Fold change (DZA/mock) of each protein was calculated for input and pellet fractions and r-proteins (pink) were compared to all other proteins (grey). ***, p-value<0.0001 (Wilcoxon rank-sum test). (**H**) List of r-proteins that assemble in the nucleus and cytosol (*Woolford and Baserga, 2013*) and ribosome biogenesis factors with greatest increase in abundance in the pellet fraction (>1.5 fold in two biological replicates) detected in DZA-treated cells by mass spectrometry (data as in 4G). See *Supplementary file 6* for full dataset.

DOI: https://doi.org/10.7554/eLife.43002.012

The following figure supplements are available for figure 4:

**Figure supplement 1.** Aggregation of orphan r-proteins during RPAS.

DOI: https://doi.org/10.7554/eLife.43002.013

*Figure 4 continued on next page*

*Figure 4 continued*

**Figure supplement 2.** Gene ontology analysis of top aggregating proteins in DZA-treated cells detected by mass spectrometry.
DOI: https://doi.org/10.7554/eLife.43002.014

r-proteins, indicating that they were not in RNA- or DNA-dependent assemblies (*Figure 4—figure supplement 1B*). To compare these results with the behavior of mature, assembled r-proteins, we grew V5-tagged Rpl10 and Rpl3 strains continuously for 5 hr in estradiol prior to DZA treatment. Under these conditions, most of the tagged r-proteins should reside in mature ribosomes, with a small fraction existing unassembled. After DZA treatment, only a modest amount of tagged r-proteins were present in the pellet, likely due to the small unassembled fraction (*Figure 4E,F*). We performed quantitative mass spectrometry to test the generality of r-protein aggregation during RPAS, and found that a broad complement of r-proteins accumulate in aggregates following DZA treatment (*Figure 4G*). Despite observing 3–5-fold increases of newly synthesized Rps26, Rpl3, and Rpl10 in the aggregate fraction following DZA treatment (*Figure 4F*), none of these proteins were in the highest ranking aggregating proteins in the mass spectrometry data (*Figure 4H*). As the mass spectrometry data are not specifically assaying newly synthesized proteins, the fold increase in aggregation is likely an underestimate, which would explain the discrepancy. Nevertheless, we observed a clear and general shift of r-proteins to the aggregate fraction following DZA treatment, beyond those that are directly downstream of Drg1 (the target of DZA) function in the cytosol (*Figure 4G, H*). Together, we conclude that RPAS results in specific aggregation of orphan r-proteins.

## RPAS disrupts nuclear and cytosolic proteostasis

In addition to finding r-proteins, particularly those that are in the large 60S subunit, amongst the strongest aggregators in DZA, we found a prominent group of nucleolar ribosome biogenesis factors (*Figure 4H*, *Figure 4—figure supplement 2*). This group contained 17 proteins, including 66S (pre-60S) associated factors such as Nop53, Nsa2, Mak16, and Cic1. Intriguingly, a number of factors involved in rRNA processing were found to be strong aggregators in DZA, including four of the components of the nuclear exosome: Lrp1, Rrp41, Rrp43, and the catalytic Rrp6. These data suggest that, in addition to causing aggregation of r-proteins downstream of Drg1 function in the cytosol, DZA treatment leads to aggregation of r-proteins assembled in the nucleus and collateral aggregation of nucleolar ribosome biogenesis factors (*Figure 4F,G,H*).

Misfolded and aggregated proteins in the cell are often toxic and have the potential to sequester proteins with essential cellular activities (*Gsponer and Babu, 2012*; *Holmes et al., 2014*; *Stefani and Dobson, 2003*). Accordingly, in addition to upregulating proteostasis factors, cells utilize spatial quality control mechanisms to minimize the deleterious effects of aggregates. For example, cells triage proteins into cytosolic aggregate depots, referred to as Q-bodies or CytoQ, where the Hsp40/70 chaperones and Hsp104 disaggregase collaborate to resolve and refold misfolded proteins (*Hill et al., 2017*; *Kaganovich et al., 2008*). Aggregates also form in the nucleus, in the intranuclear quality control compartment (INQ), which is thought to be involved in their degradation (*Hill et al., 2017*; *Miller et al., 2015a*, *Miller et al., 2015b*).

We used confocal fluorescence microscopy to follow the localization of the Hsp70 co-chaperone Sis1, which recognizes substrates and participates in nuclear aggregation and degradation (*Malinovska et al., 2012*; *Park et al., 2013*; *Summers et al., 2013*). In normal growing populations, Sis1-YFP was distributed evenly throughout the nucleus except in the nucleoli; the nucleolar protein Cfi1-mKate, which localized at the periphery of the nucleus, exhibited little or no colocalization with Sis1. Upon treatment with DZA, Sis1 drastically relocalized within the nucleus, moving to the nuclear periphery, where it formed a ring-like structure (*Figure 5A–C*). At the same time, Cfi1 relocalized from the periphery toward the middle of the nucleus, adjacent to the Sis1 ring structure. The effect of DZA on Sis1 and Cfi1 was completely blocked by inhibiting translation with CHX, consistent with the idea that newly synthesized orphan r-proteins drove the response. The subnuclear relocalization of Sis1 in response to RPAS is consistent with a role in the INQ, although the ring-like structure is distinct from the single subnuclear puncta observed following heat shock (*Malinovska et al., 2012*). In addition, we analyzed the localization of the disaggregase Hsp104, which colocalizes with aggregates and resolves them, including in a variety of proteotoxic stresses (*Glover and Lindquist, 1998*;

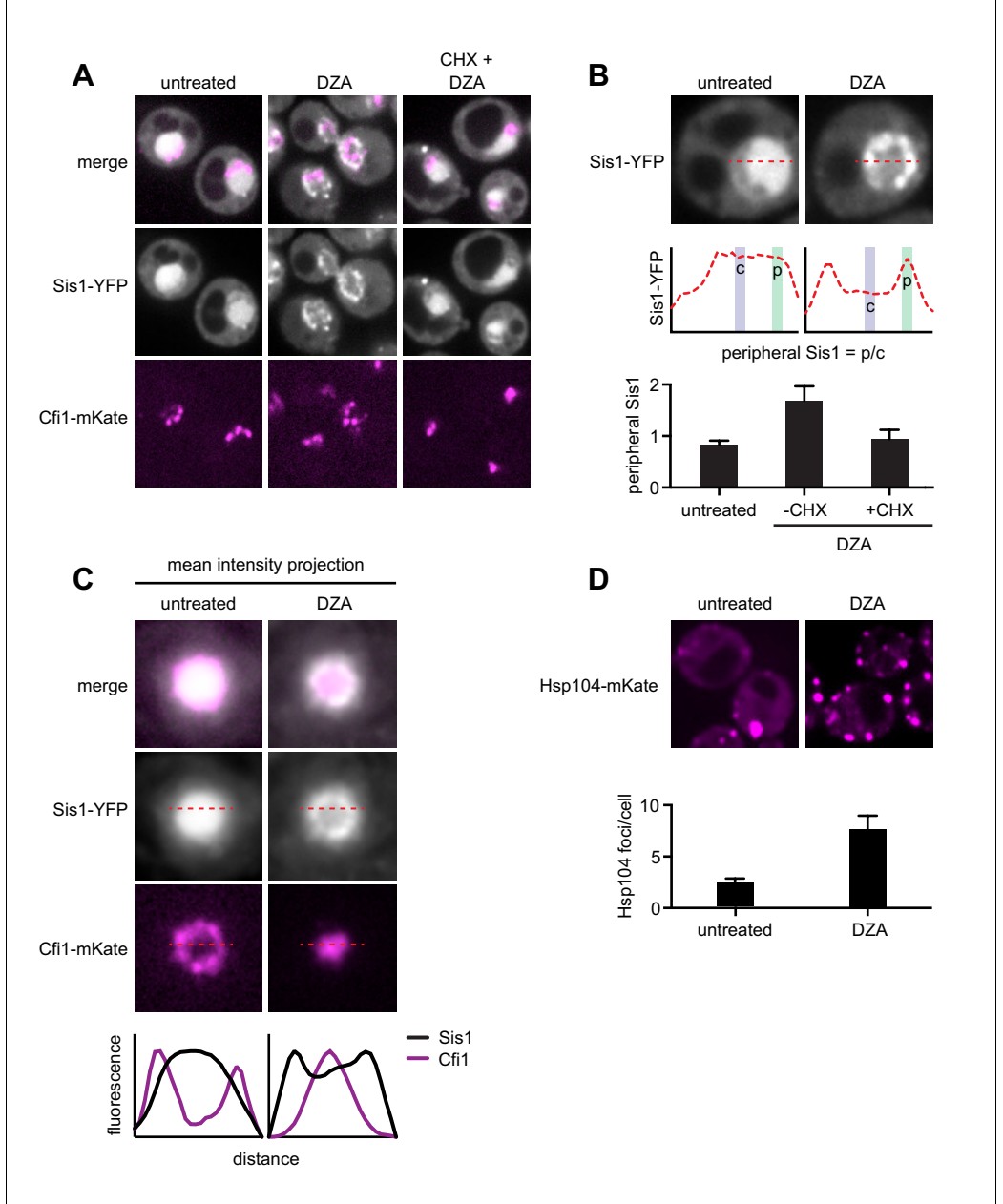

**Figure 5.** RPAS disrupts nuclear and cytosolic proteostasis. (**A**) Fluorescence micrographs of cells expressing Sis1-YFP and the nucleolar marker Cfi1-mKate after treatment with DZA (5 μg/ml, 30 min) with or without pre-treatment with CHX (200 μg/ml, 5 min). (**B**) Quantification of Sis1 relocalization to the nuclear periphery was done via fluorescence line scans and computed as the ratio of Sis1 signal at the periphery (**p**) versus the center (**c**) of the nucleus (n > 30 cells per condition). (**C**) Image segments (50 pixels) centered on the middle of the nucleus were extracted in both the Sis1-YFP and Cfi1-mKate channels for individual cells (n = 25 cells for both conditions). Images were stacked and average intensity was projected. The Cfi1 ring under control conditions results from the composite of images: in most cells it appears localized to one side, but always at the periphery of the nucleus. Fluorescent line scans quantify the localization patterns. (**D**) Micrographs of cells expressing Hsp104-mKate were imaged live in untreated conditions or after DZA treatment (5 μg/ml, 30 min). Below micrographs, quantification of number of Hsp104 foci and Sis1 peripheral localization (n > 30 cells/condition).
DOI: https://doi.org/10.7554/eLife.43002.015

*Kaganovich et al., 2008*; *Tkach and Glover, 2004*; *Zhou et al., 2014*). Untreated cells contained one or two Hsp104 foci. Treatment with DZA increased the number of cytosolic Hsp104 foci, to seven or eight per cell, likely reflecting CytoQ body formation in response to orphan r-proteins (*Figure 5D*). Based on these data, we conclude that the orphan r-proteins produced as a result of DZA treatment disrupt proteostasis in the cytosol and the nucleus.

## Hsf1 and Rpn4 support cell fitness under RPAS

To determine the physiological relevance of Hsf1 activation in response to RPAS, we tested the fitness of *hsf1* mutants and deletions of single Hsf1-dependent genes in DZA. Because *HSF1* is an essential gene, we studied a hyperphosphorylated mutant of Hsf1, *hsf1 po4\**, in which all serines are replaced with phospho-mimetic aspartates; this strain grows normally in basal conditions but is a hypoinducer of Hsf1 target genes under heat shock and has a tight temperature-sensitive growth defect (*Zheng et al., 2016*). We found that *hsf1 po4\** cells grew at wild-type rates at 30°C but were very sick under proteotoxic conditions (AZC or 37°C), demonstrating that the *hsf1 po4\** allele lacks the ability to cope with proteotoxic stress (*Figure 6A*). *hsf1 po4\** were nearly incapable of growth in DZA (*Figure 6B*), highlighting the critical role of wild-type Hsf1 in the adaptation to RPAS.

To identify which Hsf1 targets are critical for RPAS adaptation, we investigated the fitness consequence of loss of single Hsf1-dependent genes. In this analysis, we focused on genes whose loss in basal conditions is minimally perturbing but are likely to have important functions in coping with proteotoxic stress. In particular, we deleted factors involved in aggregate formation and dissolution (*HSP104*, *BTN2*, *HSP42*, *HSP26*) and proteasome-mediated degradation (*RPN4*, *TMC1*, *PRE9*); in addition, we deleted the Hsf1-independent gene *HSP12* as a negative control. Because many of these single-gene deletions do not have gross phenotypes, we used a competitive fitness assay to sensitively detect small differences in cell fitness (*Breslow et al., 2008*; *Wang et al., 2015*). Individual deletion strains expressing mCherry (mCh) were co-cultured with a wild-type reference strain expressing YFP without treatment (YPD), at 37°C, in 5 mM AZC, DMSO (vehicle), or in 15 or 30 µg/ml DZA. Competitions were maintained over the course of 5 days, and the relative proportion of wild-type and mutant cells was monitored by flow cytometry (*Figure 6C*). Deletion of most factors had no effect on fitness under any condition tested, likely due to redundancy in the mechanisms responsible for restoring proteostasis (*Figure 6—figure supplement 1*). However, loss of the transcription factor *RPN4*, which controls the basal and stress-induced levels of the proteasome (*Fleming et al., 2002*; *Wang et al., 2008*), conferred a substantial growth defect in the presence of DZA (~25 fold more severe than in the absence of drug on day 3), at 37°C, and in the presence of AZC (*Figure 6D*), suggesting that the proteasome plays a critical role in the response to RPAS. We also found that loss of the only non-essential proteasome subunit, *PRE9*, made cells DZA-resistant (*Figure 6—figure supplement 1*). Resistance to some proteotoxic stressors has been observed in weak proteasome mutants, such as *pre9*, and may be the result of compensation by alternate proteasome subunits or elevated basal levels of other proteostasis factors in this mutant (*Acosta-Alvear et al., 2015*; *Brandman et al., 2012*; *Kusmierczyk et al., 2008*; *Tsvetkov et al., 2015*). As with DZA, *rpn4* and *pre9* cells are sensitive and resistant, respectively, to endoplasmic reticulum (ER) folding stress, which involves clearance of misfolded ER proteins by the proteasome (*Kapitzky et al., 2010*; *Wang et al., 2010*). In sum, these data demonstrate that Hsf1 and its target Rpn4, which controls proteasome abundance, support cellular fitness under RPAS.

## Proteostatic strain contributes to the growth defect of cells under RPAS

We hypothesized that the proteotoxic stress created by orphan r-proteins contributes to the growth defect of cells under RPAS beyond what would be expected from the effects of a reduced ribosome pool. Because Hsf1 responds to and is required for growth under RPAS, we uncoupled Hsf1 from the proteostasis network and placed it under exogenous control to test whether enhanced proteostasis would modulate the DZA-induced growth defect. For this purpose, we placed a chimeric fusion of the Hsf1 DNA-binding domain with the transactivation domain VP16 (Hsf1[DBD]-VP16) under the control of an estradiol-responsive promoter in a strain lacking wild-type *HSF1*, allowing exogenous upregulation of the Hsf1 regulon by addition of estradiol. The Hsf1[DBD]-VP16 strain was more sensitive to DZA than the wild-type strain, further supporting the importance of wild-type *HSF1* in

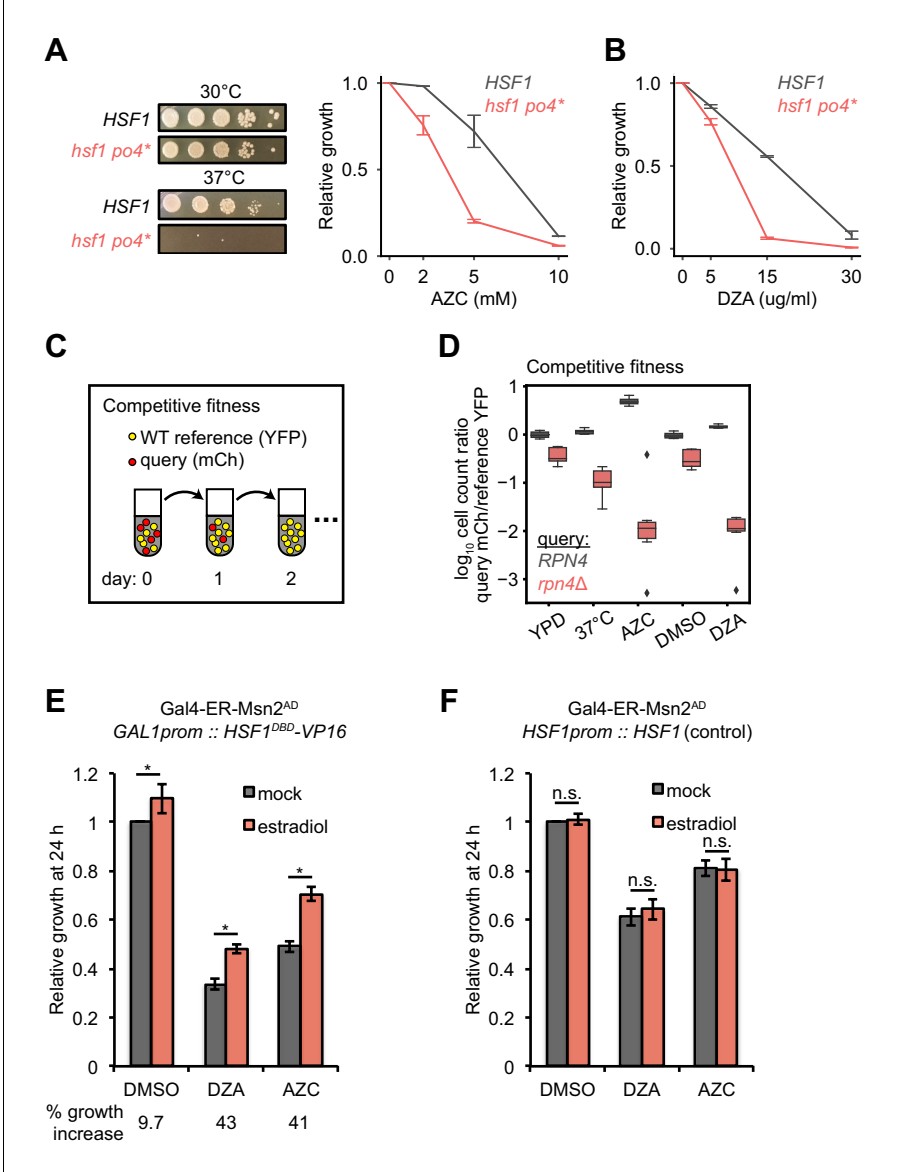

**Figure 6.** Hsf1 and Rpn4 support cell fitness under RPAS. (**A**) Growth defects of *hsf1 po4** cells. Left panels, wild-type (*HSF1)* and mutant (*hsf1 po4**, all serine to aspartate) cells were serially diluted 1:10 onto YPD plates and incubated at 30 or 37°C for 2 days. Right panel, cells were grown for 24 hr in the presence of the indicated concentration of AZC and relative growth (compared to untreated) was determined by $OD_{600}$. Line represents the average and error bars the range of n = 2 biological replicates. (**B**) Cells were grown for 24 hr in the presence of the indicated concentration of DZA and relative growth (compared to untreated) was determined by $OD_{600}$. Line represents the average and error bars the range of n = 2 biological replicates. (**C**) Schematic of competitive fitness assay. Wild-type (WT) cells expressing YFP and query cells expressing mCherry (mCh) were co-cultured in each condition over 5 days. Abundance of YFP +and mCh +cells was determined daily by flow cytometry. (**D**) The $\log_{10}$ ratio of mCh+ (query) to YFP+ (WT reference) of wild-type (*RPN4*) and *rpn4Δ* cells after 3 days of co-culture in YPD, YPD at 37°C, 5 mM AZC, vehicle (DMSO) and DZA (15 µg/ml). Box plot of n = 8 biological replicates with outliers shown as diamonds. (**E**) Growth of cells expressing a synthetic Hsf1 construct severed from negative regulation by chaperones (Hsf1$^{DBD}$-VP16) was expressed under an estradiol-responsive promoter. Pre-conditioning was performed with estradiol (2 nM) for 3 hr prior to addition of DMSO, DZA (8 µg/ml), or AZC (2.5 mM) for an additional 21 hr. Growth was determined as $OD_{600}$ normalized to DMSO control. Bar height depicts the average and error bars the standard deviation of n = 3 biological replicates. Values below indicate the average % increase in growth by estradiol pre-conditioning versus mock. *, all p<0.01 (Student's t-test). (**F**) Results of experiments performed identically as described in A, but with an isogenic strain containing *HSF1* under its WT promoter

*Figure 6 continued on next page*

*Figure 6 continued*

instead of the Hsf1$^{DBD}$-VP16 under an estradiol-responsive promoter. n.s., not significant, all p>0.1 (Student's t-test).

DOI: https://doi.org/10.7554/eLife.43002.016

The following figure supplements are available for figure 6:

**Figure supplement 1.** Competitive fitness of strains lacking single Hsf1-dependent genes.

DOI: https://doi.org/10.7554/eLife.43002.017

**Figure supplement 2.** Growth improvement is not due to changes in cell size.

DOI: https://doi.org/10.7554/eLife.43002.018

---

the RPAS response (*Figure 6E,F*). To determine whether upregulation of the Hsf1 regulon alleviates the DZA growth defect, we pre-conditioned cells with a 3 hr estradiol treatment, and then measured cell growth after 21 hr of exposure to DZA, AZC, or DMSO (vehicle). Pretreatment with estradiol yielded a > 40% growth enhancement in DZA that was independent of changes to cell size. Similar effects were observed after growth in AZC, which induces global proteotoxicity, whereas only a 9% growth rate increase was observed for vehicle-treated cells (*Figure 6E,F* and *Figure 6—figure supplement 2*). These data suggest that the proteotoxic stress of RPAS slows growth, which can be rescued by exogenous amplification of the proteostasis network.

## Cells producing fewer ribosomes show reduced proteostatic strain in RPAS

Our data demonstrate that rapidly proliferating yeast cells experience an acute loss of proteostasis when ribosome assembly is disrupted. We asked whether cells producing fewer ribosomes would experience an attenuated proteotoxic stress during RPAS. To this end, we analyzed wild-type yeast grown in rich medium containing the optimal carbon source glucose or the suboptimal (respiratory) carbon source glycerol (*Metzl-Raz et al., 2017*). Under these conditions, cells doubled every 1.6 and 3.7 hr, respectively. When challenged with DZA, cells grown in glycerol demonstrated a lower level of Hsf1 target gene activation (*Figure 7A*). To analyze the impact of reduced ribosome biogenesis without changing the carbon source, we analyzed cells lacking the gene *SCH9*, whose product controls ribosome production at the transcriptional level, in glucose-containing medium. As with wild-type cells in glycerol, *sch9Δ* cells showed lower levels of Hsf1 target gene activation by DZA (*Figure 7B*). Importantly, we observed that DZA treatment altered the processing of rRNA under all conditions (*Figure 7—figure supplement 1*), validating that ribosome assembly was being disrupted. Thus, the proteotoxic strain was stronger in cells with higher rates of ribosome production, indicating that proliferating cells are at a stronger risk of experiencing RPAS.

## Discussion

Here, we report an extraribosomal consequence of aberrant ribosome assembly: collapse of proteostasis resolved by an Hsf1-dependent response. We propose a model wherein excess orphan r-proteins that arise from aberrations in ribosome biogenesis drive proteotoxicity and impact cellular fitness under r-protein assembly stress (*Figure 7C*). In turn, the master proteostasis transcription factor Hsf1 is activated to increase the abundance of folding and degradation machineries, likely following sequestration of chaperones such as Hsp40 and Hsp70 by r-protein aggregates (*Zheng et al., 2016*). The proteostatic response supports cell fitness and is capable of protecting cells from r-protein assembly stress. Thus, proliferating cells accept a tradeoff between the risk of proteotoxicity and the growth benefits of high ribosome production. The resulting balancing act is vulnerable to disruption by a variety of genetic and chemical insults, necessitating protective mechanisms capable of restoring the balance. Interestingly, several r-proteins are produced in excess, for instance in human tissue culture cells, and are rapidly targeted for degradation by the ubiquitin-proteasome system (*Abovich et al., 1985*; *Lam et al., 2007*; *McShane et al., 2016*; *Sung et al., 2016a*, *Sung et al., 2016b*). We therefore propose that in the perturbations modeled in this work, cells are challenged with a larger proportion of orphan r-proteins that overwhelms the canonical clearance mechanisms, necessitating an increase in proteostasis capacity, consistent with the importance of both Hsf1 and Rpn4 in RPAS (*Figure 6B,D*).

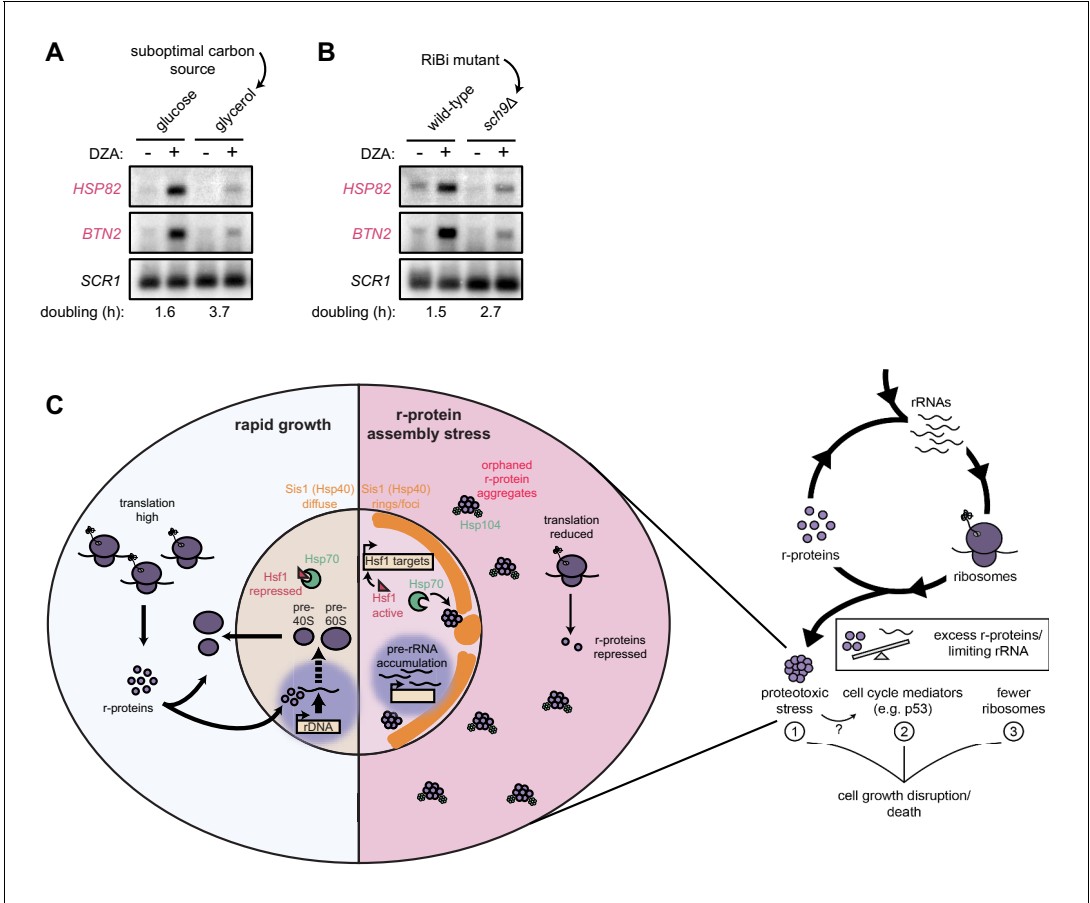

**Figure 7.** Cells producing fewer ribosomes show reduced proteostatic strain in RPAS. (**A**) Wild-type cells were grown to mid-log in rich medium with either 2% glucose or glycerol and treated with DMSO (vehicle, -) or DZA (+) for 15 min. Shown are Northern blots for Hsf1 target genes *HSP82 and BTN2*. (**B**) Wild-type and *sch9Δ* cells were both grown to mid-log in rich medium with 2% glucose and treated with DMSO (vehicle, -) or DZA (+) for 15 min. Shown are Northern blots for Hsf1 target genes *HSP82 and BTN2*. (**C**) Model of how disruptions to ribosome biogenesis leads to RPAS and the impacts on cellular physiology. During proliferation, cells rapidly produce ribosomes through coordinated synthesis of r-proteins (purple circles) in the cytoplasm and rRNAs in the nucleolus. Perturbations that result in orphan r-proteins result in proteotoxic stress following r-protein aggregation (left panel). In the cytoplasm, aggregates are visible via Hsp104 foci and translation is downregulated. In the nucleus, Hsp40 Sis1 (orange), and possibly Hsp70, are targeted to aggregates and the nucleolus moves from the nuclear periphery, to adjacent to Sis1-marked 'rings'. Concomitantly, pre-rRNA accumulates, r-protein genes are transcriptionally repressed, and Hsf1 is liberated from Hsp70 sequestration to activate target genes encoding protein folding and degradation machinery. Proteostasis collapse stalls growth independently from reduced pools of ribosomes (right panel).
DOI: https://doi.org/10.7554/eLife.43002.019

The following figure supplement is available for figure 7:

**Figure supplement 1.** Disrupted rRNA processing in DZA-treated cells.
DOI: https://doi.org/10.7554/eLife.43002.020

It is possible that, rather than aggregated r-proteins, pre-40S/60S precursors accumulated in the nucleolus elicit RPAS. Though we cannot definitively test this alternative model, we find it unlikely for several reasons. First, many lines of evidence point towards Hsf1 activation requiring accumulation of misfolded/aggregated proteins that titrate chaperones away from binding and inactivating Hsf1 (*Shi et al., 1998*; *Zheng et al., 2016*), making it difficult to envision a model wherein precursors per se drive Hsf1 activation independent of r-protein aggregation. Second, the RPAS response is also activated by depletion of rRNA processing factors, which remove the platform (rRNA) for precursor assembly altogether. Third, in the case of DZA treatment, we found many additional r-proteins that aggregate beyond those that are downstream of Drg1 function, including many that assemble at the earliest stages of precursor formation in the nucleolus. Thus, we favor a model

wherein aberrations in ribosome biogenesis that affect both rRNA production and r-protein assembly lead to RPAS due to aggregation of orphan r-proteins in the nucleus and cytosol.

Given the conservation of proteostasis mechanisms and ribosome biogenesis, we suspect that disrupted ribosome assembly might also cause proteotoxic stress in other eukaryotes. Certainly, many conditions have the potential to orphan r-proteins, thereby straining proteostasis. For example, DNA-damaging chemotherapeutic agents like etoposide, camptothecin, and 5-fluorouracil and transcription inhibitors like actinomycin D disrupt the nucleolus and rRNA processing (*Burger et al., 2010*). Indeed, several Hsf1 targets are seen upregulated by and may be important in responding to DNA damaging agents (*Miller et al., 2015a*; *Tkach et al., 2012*; *Workman et al., 2006*). Environmental stressors such as heat shock also deform the nucleolus, and many other stressors in yeast cause accumulation of pre-rRNA (*Boulon et al., 2010*; *Kos-Braun et al., 2017*). Imbalanced production of r-proteins arises in mutations found in ribosomopathies, as well as in aging (*David et al., 2010*) and cancer (*Guimaraes and Zavolan, 2016*). Because ribosome biogenesis is not a constitutive process, but instead fluctuates in response to nutrient availability, stress, cell growth, and differentiation cues (*Lempiäinen and Shore, 2009*; *Mayer and Grummt, 2006*), these conditions are likely to acutely challenge ribosome biogenesis and lead to periodic disruptions to proteostasis. The severity of the resulting phenotype may relate to cell growth rate and the required level of ribosome production in a cell type/state (*Figure 7A,B*), which suggests a possible mechanism for why certain cell types are especially vulnerable to disrupted ribosome biogenesis, such as in ribosomopathies.

Proteotoxic stress has been extensively linked to overall disruption of cellular homeostasis (*Gsponer and Babu, 2012*; *Holmes et al., 2014*; *Stefani and Dobson, 2003*). While the molecular basis for how protein aggregates compromise cell health is not fully understood, one demonstrated possibility is that aggregates sequester other proteins with essential functions (*Olzscha et al., 2011*). Thus, the proteotoxic stress elicited by RPAS has the potential to severely disrupt cellular homeostasis, consistent with our findings that alleviating proteotoxic stress enhances cell growth under RPAS (*Figure 6E*). Differences among cell types in the ability to withstand proteotoxic conditions might contribute to the phenotypic variability in response to ribosome assembly defects.

The gene expression response mounted by cells experiencing RPAS provides clues regarding how the cell deals with toxic orphan r-proteins. The requirement for an Hsf1-mediated response suggests that upregulation of the folding and/or degradation machinery contributes to this resolution. The extreme sensitivity of *rpn4* cells to RPAS suggests an important role for proteasome-mediated degradation of orphan r-proteins. Consistent with this, yeast and human cells degrade r-proteins produced in excess, and cells lacking this quality control mechanism contain aggregated r-proteins (*McShane et al., 2016*; *Sung et al., 2016a*, *Sung et al., 2016b*). Indeed, the proteotoxicity of excess r-proteins may explain why cells evolved mechanisms to prevent their accumulation above stoichiometric levels, even in aneuploid cells (*Dephoure et al., 2014*).

Activation of the Hsf1 regulon in RPAS is the consequence of newly synthesized r-proteins that cannot reach their normal destination and therefore fail to assemble into a cognate complex, leading to their aggregation. Similarly, the mitochondrial unfolded protein response is activated when assembly of mitochondrial complexes is disrupted (*Yoneda et al., 2004*). Blocking import of organellar proteins into the ER or mitochondria results in cytosolic proteotoxic stress (*Brandman et al., 2012*; *Wang et al., 2014*; *Weidberg and Amon, 2018*; *Wrobel et al., 2015*). Thus, aberrant accumulation of orphan proteins – that is, those that do not arrive at their appropriate complex or subcellular location – is a hallmark of proteostasis loss, which is resolved by pathways tailored for each cellular compartment. Given that the nucleolus is morphologically disrupted and recruits chaperones such as Hsp70 under stress, including heat shock and proteasome inhibition (*Lam et al., 2007*; *Liu et al., 1996*; *Pelham, 1984*), it is tempting to speculate that RPAS is responsible, at least in part, for Hsf1 activation in response to various stress stimuli. Consistently, new r-proteins undergo ubiquitination, localize in protein aggregates, and associate with chaperones under heat shock (*Fang et al., 2014*; *Ruan et al., 2017*; *Shalgi et al., 2013*). R-proteins, due to their exceptionally high abundance, complex assembly pathway, and aggregation-prone nature, represent a particularly vulnerable group of proteins.

Particular cell types and cell states, such as tumor cells or differentiating erythropoietic precursors, have exceptional demand for high ribosome production (*Mills and Green, 2017*; *Pelletier et al., 2018*). Intriguingly, both of these cell states are unusually sensitive to disruption of proteostasis. Erythroid differentiation is highly reliant on Hsp70 availability, as evidenced by the fact

that Hsp70 sequestration can result in the anemic phenotype of beta-thalassemia (*Arlet et al., 2014*). Similarly, cancer cells are sensitized to small molecules that dampen the proteostasis network (*Balch et al., 2008*; *Joshi et al., 2018*). In this work, we showed that exogenous activation of the Hsf1 regulon protects yeast from RPAS. Future studies should seek to determine whether an analogous strategy can therapeutically mitigate phenotypes of disrupted ribosome biogenesis in disease processes.

# Materials and methods

## Key resources table

| Reagent type (species) or resource | Designation | Source or reference | Identifiers | Additional information |
|---|---|---|---|---|
| Antibody | Mouse monoclonal anti-V5 | Invitrogen | Invitrogen: R960-25; RRID:AB_2556564 | 1:2000 |
| Antibody | Mouse monoclonal anti-Pgk1 | Abcam | Abcam: ab113687; RRID:AB_10861977 | 1:10,000 |
| Antibody | Rabbit polyclonal anti-Rpb1 | Santa Cruz Biotechnology | Santa Cruz:sc-25758; RRID:AB_655813 | 1:1000 |
| Antibody | Rabbit polyclonal anti-Hsp104 | Enzo Life Sciences | Enzo:ADI-SPA-1040 | 1:1000 |
| Antibody | Mouse monoclonal anti-Hsp70 | Abcam | Abcam: ab5439; RRID:AB_304888 | 1:1000 |
| Antibody | Rabbit monoclonal anti-phos-Rps6 (Ser235/236) | Cell Signaling Technology | Cell Signaling: 4858; RRID:AB_916156 | 1:2000 |
| Antibody | Rabbit polyclonal anti-phos-eIF2α (Ser51) | Invitrogen | Invitrogen: 44–728G; RRID:AB_2533736 | 1:1000 |
| Chemical compound, drug | auxin (indole-3-acetic acid) | Sigma-Aldrich | Sigma-Aldrich: I3750 | |
| Chemical compound, drug | diazaborine | Millipore Sigma | Millipore Sigma:530729 | |
| Chemical compound, drug | cycloheximide | Sigma-Aldrich | Sigma-Aldrich: C4859 | |
| Chemical compound, drug | L-azetidine-2-carboxylic acid | Sigma-Aldrich | Sigma-Aldrich: A0760 | |
| Chemical compound, drug | diamide | Sigma-Aldrich | Sigma-Aldrich: D3648 | |
| Chemical compound, drug | rapamycin | LC Labs | LC Labs:R-5000 | |
| Chemical compound, drug | beta-estradiol | Sigma-Aldrich | Sigma-Aldrich: E2758 | |

## Yeast strain construction and growth

Strains were constructed by standard transformation techniques (*Gietz and Schiestl, 2007*). Gene tagging and deletion was carried out using PCR products or integrating plasmids, and transformants were verified by colony PCR and western blotting where relevant. The Hsf1 activity reporters contain four Hsf1-binding sites (heat shock element, HSE) upstream of a reporter gene (*Brandman et al.,*

*2012*; *Zheng et al., 2016*). The *HSE-GFP* and *HSE-mVenus* reporters were integrated at *URA3* and *LEU2*, respectively, and were used interchangeably depending on experimental requirements. *Os*TIR1 driven by the *GPD1* promoter was integrated at *LEU2*. The AID tag was added to a *TIR1*-containing strain by transformation with the *V5-IAA7::KANMX6* cassette. Further transformation of AID strains often resulted in loss of *Os*TIR1 activity, reflected by failure to deplete the tagged protein in auxin; accordingly, such transformations were not performed. The *DRG1* and *DRG1*$^{V725E}$ strains were constructed in a diploid by deletion of one *DRG1* allele followed by transformation with the WT or mutant allele on a *URA3*-marked CEN/ARS plasmid (see 'Cloning'). Clones containing only the plasmid-borne copy were isolated by sporulation and tetrad dissection. Estradiol-inducible expression strains were generated with a plasmid containing the V5-tagged ORF downstream of the *GAL1* promoter that integrates at *HIS3* in a background expressing the Gal4-ER-Msn2$^{AD}$ transcription factor (Stewart-Ornstein et al., 2012). All strains and plasmids are listed in *Supplementary files 1* and *2*, respectively.

All experiments were performed at 30°C with cultures were grown in standard YPD (1% yeast extract, 2% peptone, 2% dextrose, pH 5.5) medium unless indicated otherwise. Where indicated, SCD (0.2% synthetic complete amino acids [Sunrise], 0.5% ammonium sulfate, 0.17% yeast nitrogen base, 2% dextrose, pH 5.5) medium was used. Heat shock was performed by adding an equal volume of 44°C media to the culture and immediately shifting to a 37°C incubator.

## Drug treatments

Treatments were generally carried out in log-phase cultures at OD ~0.4–0.6, depending on the length of treatment, such that cultures remained in log growth during the course of the experiment. For drugs dissolved in DMSO, vehicle-only controls contained the same final volume of DMSO. Auxin (indole-3-acetic acid, Sigma-Aldrich) was prepared fresh daily at 100 mM in ethanol and added at a final concentration of 100 µM. Diazaborine (DZA, Calbiochem) was prepared at 15 mg/ml in DMSO (stored at −20°C, protected from light) and used at the indicated concentration. Cycloheximide (Sigma-Aldrich) was purchased as a 100 mg/ml DMSO stock and added at a final concentration of 100 µg/ml (for sucrose gradients) or 200 µg/ml (for stress experiments). AZC (L-azetidine-2-carboxylic acid, Sigma-Aldrich) was prepared at 1 M in water and used at the indicated concentration. Diamide (Sigma-Aldrich) was prepared at 1 M in water and added at a final concentration of 1.5 mM. Rapamycin (LC Laboratories) was prepared fresh daily in ethanol and used at a final concentration of 200 ng/ml (to inhibit r-protein synthesis) or 1 µg/ml (for anchor-away, in a rapamycin-resistant *tor1-1* background). Beta-estradiol (Sigma-Aldrich) was prepared as a 1000X stock for each experiment in ethanol and added to the indicated final concentration.

## Cloning

*DRG1*, including promoter and terminator regions, was PCR amplified from genomic DNA with tails containing *Bam*HI and *Not*I sites and cloned into pBluescript KS. The *DRG1*$^{V725E}$ mutant was constructed by Q5 site-directed mutagenesis. WT and mutant were subcloned using the same restriction sites into pRS316 (*URA3* CEN/ARS) and verified by sequencing of the full insert. V5-tagged ORFs were ordered as gBlocks (IDT) with a C-terminal 6xGly-V5 tag and *Xho*I and *Not*I sites and cloned into pNH603 under the *GAL1* promoter. RP ORFs had the sequence of the genomic locus and *GFP* encoded enhanced monomeric GFP (F64L, S65T, A206K).

## Total protein extraction and western blotting

Each western blot was performed with a minimum of two biological replicates unless otherwise stated and a representative blot is shown. Protein extraction was adapted from the alkaline lysis method (*Kushnirov, 2000*). One milliliter of a mid-log culture was harvested in a microfuge, aspirated to remove supernatant, and snap-frozen on liquid nitrogen. Pellets were resuspended at RT in 50 µl 100 mM NaOH. After 3 min, 50 µl 2X SDS buffer (4% SDS, 200 mM DTT, 100 mM Tris pH 7.0, 20% glycerol) was added, and the cells were lysed on a heat block for 3 min at 95°C. Cell debris was cleared by centrifugation at 20,000 *g* for 5 min.

Extracts were resolved on NuPAGE Bis-Tris gels (Invitrogen), transferred to nitrocellulose on a Trans-Blot Turbo (Bio-Rad), and blocked in 5% milk/TBST (0.1% Tween-20). AID-tagged and V5-tagged proteins were detected with mouse anti-V5 (Invitrogen, R960-25, 1:2000). Pgk1 was

detected using mouse anti-Pgk1 (Abcam, ab113687, 1:10,000). Rpb1 was detected with rabbit anti-Rpb1 (y-80, Santa Cruz Biotechnology, sc-25758, 1:1,000). Hsp104 was detected with rabbit anti-Hsp104 (Enzo Life Sciences, ADI-SPA-1040, 1:1000). Hsp70 was detected with mouse anti-Hsp70 (3A3, Abcam, ab5439, 1:1000). Rps6 phosphorylated at Ser235/236 was detected with rabbit anti-phos-Rps6 (D57.2.2E, Cell Signaling Technology, 1:2,000). eIF2$\alpha$ phosphorylated at Ser51 was detected with rabbit anti-phos-eIF2$\alpha$ (Invitrogen, 44–728G, 1:1000). Pgk1 and Rpb1 were used as loading controls. Cy3-labeled secondary antibodies were used, and immunoreactive bands were imaged on a Typhoon.

## Proteomics

Samples were prepared essentially as previously described (*Gupta et al., 2018*; *Sonnett et al., 2018b*). Soluble (input) and pelleting proteins were extracted exactly as in section 'Protein aggregation assay.' About 200 µg of protein were cleaned with a chloroform/methanol precipitation (*Wessel and Flügge, 1984*). Proteins were resuspended in 6 M GuHCl, diluted to 2 M GuHCl with 10 mM EPPS at pH = 8.5, and digested with 10 ng/µL LysC (Wako) at 37°C overnight. Samples were further diluted to 0.5 M GuHCl and digested with an additional 10 ng/µL LysC and 20 ng/µL sequencing grade Trypsin (Promega) at 37°C for 16 hr. TMT tagging, and peptide desalting by stage-tipping was performed as previously described (*Gupta et al., 2018*; *Sonnett et al., 2018b*). LC-MS. LC-MS experiments were performed on a Thermo Fusion Lumos equipped with an EASY-nLC 1200 System HPLC and autosampler (Thermo). During each individual run, peptides were separated on a 100–360 µm inner-outer diameter microcapillary column, which was manually packed in house first with ~0.5 cm of magic C4 resin (5 µm, 200 Å, Michrom Bioresources) followed by ~30 cm of 1.7 µm diameter, 130 Å pore size, Bridged Ethylene Hybrid C18 particles (Waters). The column was kept at 60°C with an in house fabricated column heater (*Richards et al., 2015*). Separation was achieved by applying a 6–30% gradient of acetonitrile in 0.125% formic acid and 2% DMSO at a flow rate of ~350 nL/min over 90 min for reverse phase fractionated samples. A voltage of 2.6 kV was applied through a PEEK microtee at the inlet of the column to achieve electrospray ionization. The data were acquired with a MultiNotch MS3 method essentially as previously described (*Wühr et al., 2015*). Five SPS precursors from the MS2 were used for the MS3 using MS1 isolation window sizes of 0.5 for the MS2 spectrum and isolation windows of 1.2, 1.0, and 0.8 m/z for 2+, 3 + and 4–6 + peptides, respectively. An orbitrap resolution of 50 k was used in the MS3 with an AGC target 1.5e5 and a maximum injection time of 100 ms. Proteomics data were analyzed essentially as previously described (*Sonnett et al., 2018a*). Protein-level data are presented in *Supplementary file 6*. Raw signal-to-noise measurements for each TMT channel (corresponding to one sample) were normalized but dividing each protein by the sum of all signal in that channel and multiplying by 10e6, resulting in parts per million (ppm). Gene ontology (GO) term enrichment was performed using the Saccharomyces Genome Database GO term finder tool on the 51 proteins whose input-normalized fold change in the pellet of DZA-treated cells was >1.5X in both replicates (see *Supplementary file 6*). The list of all proteins quantified in the dataset was used as the background set.

## Total RNA extraction and northern blotting

Each Northern blot was performed with a minimum of two biological replicates unless otherwise stated and a representative blot is shown. Two milliliters of a mid-log culture were harvested in a microfuge, aspirated to remove supernatant, and snap-frozen on liquid nitrogen. RNA was extracted by the hot acid-phenol method and ethanol precipitated. RNA purity and concentration were determined on a NanoDrop.

Typically 5 µl (5 µg) of RNA was mixed with 16 µl sample buffer (10 µl formamide, 3.25 µl formaldehyde, 1 µl 20X MOPS, 1 µl 6X DNA loading dye, 0.75 µl 200 µg/ml ethidium bromide) and denatured for 10 min at 65°C. After chilling briefly on ice, samples were loaded onto a 100 ml 1.2% agarose/1X MOPS gel and electrophoresed for 90 min at 100V in 1X MOPS in a Thermo EasyCast box. Some gels contained 6% formaldehyde and ran for 5 hr, but a 90 min run without formaldehyde gave sharper, more even bands. We also found that low EEO agarose gave the best results. RNA integrity and equal loading were examined by imaging ethidium bromide to visualize rRNA bands. RNA was fragmented in the gel for 20 min in 3 M NaCl/10 mM NaOH before downward capillary transfer on a TurboBlotter apparatus using the manufacturer's blotting kit. Transfer ran for 90 min in

3 M NaCl/10 mM NaOH, and then the membrane was UV crosslinked. Pre-5.8S rRNA was resolved by running 1 µg RNA (in 1X TBE-urea loading buffer) on a 6% TBE-urea gel in 0.5X TBE. RNA was electroblotted to a membrane and UV-crosslinked.

RNA was detected with either small DNA oligonucleotides or large (100–500 bp) double-stranded DNA (see *Supplementary file 3*). For oligo probes, the membrane was pre-hybridized at 42°C in ULTRAhyb-Oligo buffer (Thermo Fisher Scientific). The oligo was 5' end–labeled in a reaction containing 25 pmol oligo, 10 U T4 PNK, 2 µl gamma-$^{32}$P-ATP (PerkinElmer), and 1X PNK buffer. Probe was hybridized overnight and washed twice in 2X SSC/0.5% SDS at 42°C for 30 min before exposure on a phosphor screen and imaging on a Typhoon. For dsDNA probes, the membrane was pre-hybridized at 42°C in 7.5 ml deionized formamide, 3 ml 5 M NaCl, 3 ml 50% dextran sulfate, 1.5 ml 50X Denhardt's, 750 µl 10 mg/ml salmon sperm DNA, 750 µl 1 M Tris 7.5, 75 µl 20% SDS. Probes were made in a reaction containing 50 ng of a PCR product as template, random hexamer primers, Klenow (exo-), and 5 µl alpha-$^{32}$P-ATP (PerkinElmer). Denatured probes were hybridized overnight and washed twice in 2X SSC/0.5% SDS at 65°C for 30 min before exposure on a phosphor screen and imaging on a Typhoon scanner.

## Chromatin immunoprecipitation-quantitative PCR (ChIP-qPCR)

ChIP was performed based off of standard approaches. Fifty milliliters of a mid-log culture were crosslinked in 1% formaldehyde for 30 min at RT and quenched in 125 mM glycine for 10 min. Cells were pelleted and washed twice in ice-cold PBS before snap-freezing on liquid nitrogen. Chromatin was extracted in LB140 (50 mM HEPES pH 7.5, 140 mM NaCl, 1 mM EDTA, 1% Triton X-100, 0.1% sodium deoxycholate, 0.1% SDS and 1X protease inhibitor cocktail [cOmplete EDTA-free, Roche]) by glass bead beating. Chromatin was sonicated to 100–300 bp on a Bioruptor (Diagenode) and diluted 1:10 in WB140 (LB140 without SDS). Diluted chromatin (1.5 ml, corresponding to ~6 ml of the original cell culture volume) was incubated overnight at 4°C with 1 µl rabbit anti-Hsf1 serum (kind gift from Dr. David Gross, Louisiana State University), or normal rabbit serum as a negative control. Twenty-five microliters of washed Protein A Dynabeads (Invitrogen) were added, and the sample was incubated for 4 hr. One wash each was performed for 5 min in WB140 (140 mM NaCl), WB500 (500 mM NaCl), WBLiCl (250 mM LiCl), and TE. Samples were eluted from beads in TE/1% SDS and de-crosslinked overnight at 65°C, followed by RNase A and proteinase K treatment and cleanup on columns. Input and IP DNA were quantified using Brilliant III SYBR Green Master Mix (Agilent Technologies) in technical triplicate for each biological replicate sample. A dilution curve was generated for each input. Data are recorded for each IP as percent of input using Ct values. Primers are available in *Supplementary file 3*.

## Protein aggregation assay

Insoluble proteins were isolated using the protocol described in *Wallace et al. (2015)*. Twenty-five milliliter cultures were grown to mid-log and treated as indicated, pelleted for 1 min at 3000 *g*, and rinsed once in 1 ml ice-cold WB (20 mM HEPES pH 7.5, 120 mM KCl, 2 mM EDTA). The pellet was resuspended with 100 µl SPB and dripped into 2 ml safe-lock tubes filled with liquid nitrogen along with a 7 mm stainless steel ball (Retsch). Cells were cryogenically lysed on a Retsch Mixer Mill 400 by four cycles of 90 s at 30 Hz and re-chilled on liquid nitrogen between each cycle. The grindate was thawed with 400 µl SPB (WB +0.2 mM DTT +1X protease inhibitors [cOmplete EDTA-free, Roche] +1X phosphatase inhibitors [PhosSTOP, Sigma-Aldrich]) for 5 min on ice with repeated flicking and gentle inversion. Where indicated, 2 µl benzonase (Sigma-Aldrich) was included in SPB to degrade RNA and DNA for 10 min on ice. The lysate was clarified for 30 s at 3,000 *g* to remove cell debris. Twenty microliters of extract was reserved as input. The remaining extract was centrifuged for 20 min at 20,000 *g* to pellet insoluble proteins. The supernatant was decanted and the pellet rinsed with 400 µl ice-cold SPB with brief vortexing and centrifuged again for 20 min. The pellet was resuspended in 200 µl IPB (8 M urea, 2% SDS, 20 mM HEPES pH 7.5, 150 mM NaCl, 2 mM EDTA, 2 mM DTT, 1X protease inhibitors) at RT. The input was diluted with 160 µl water and 20 µl 100% TCA and precipitated for 10 min on ice, centrifuged for 5 min at 20,000 *g* and washed with 500 µl ice-cold acetone. Inputs were resuspended in 100 µl IPB. Input and pellet fractions were centrifuged for 5 min at 20,000 *g*, RT. Ten microliters of input (0.5%) and pellet (5%, 10X) were used for western blotting as above.

## Sucrose gradient sedimentation

Fifty-milliliter cultures were grown to mid-log and treated as indicated, followed by addition of CHX to 100 µg/ml and incubation for 2 min. All following steps were performed on ice or at 4°C. Cells were pelleted for 2 min at 3000 $g$, washed once in 10 ml buffer (20 mM Tris pH 7.0, 10 mM $MgCl_2$, 50 mM KCl, 100 µg/ml CHX), and once in 1 ml buffer. Cells were pelleted in a microfuge and snap-frozen on liquid nitrogen. Cells were lysed by addition of 400 µl glass beads and 400 µl lysis buffer (20 mM Tris pH 7.0, 10 mM $MgCl_2$, 50 mM KCl, 100 ug/ml CHX, 1 mM DTT, 50 U/ml SUPERaseIn [Thermo Fisher], 1X protease inhibitors) followed by bead beating for six cycles (1 min on, 2 min off) on ice. Lysate was clarified 10 min at 20,000 $g$. A continuous 12 ml 10–50% sucrose gradient was prepared in 20 mM Tris pH 7.0, 10 mM $MgCl_2$, 50 mM KCl, 100 µg/ml CHX on a BioComp Gradient Station, and 200 µl (~20 A260 units) lysate was layered onto the top and spun for 3 hr at 40,000 rpm in a SW41 rotor. Absorbance profiles and fractions were collected on a BioComp Gradient Station.

## Competitive fitness and growth assays

Fitness experiments were performed as described (*Wang et al., 2015*). Query strains (WT and deletions) expressing *TDH3p-mCherry* were co-cultured with a reference strain expressing *TDH3p-YFP*. All strains were inoculated from single colonies into liquid YPD and grown to saturation. Query and reference strains were mixed 1:1 (v:v) at a total dilution of 1/100 and grown for 6 hr to an $OD_{600}$ of 0.2–0.5. Co-cultured cells were diluted 1/10 to a final $OD_{600}$ of 0.02–0.05 in YPD alone or YPD with: 0.1% (v/v) DMSO (vehicle), 15 µg/mL DZA, 30 µg/mL DZA, or 5 mM AZC and grown at 30°C. Samples were also diluted in YPD and grown at 37°C. Samples were co-cultured for 5 days and diluted 1/100 into fresh media every 24 hr. At each time point, an aliquot of each sample was transferred to TE and quantified by flow cytometry on a Stratedigm S1000EX cytometer. Manual segmentation was used to identify the query and reference strain populations. Data are available in *Supplementary file 5*.

To determine relative growth of *HSF1* and *hsf1 po4\** (*Figure 6A,B*) and *DRG1* and *DRG1 V725E* (*Figure 2—figure supplement 1B*), overnight cultures were diluted to $OD_{600}$ ~0.05 in the indicated condition, grown for 24 hr, and $OD_{600}$ measured. 'Relative growth' is the $OD_{600}$ for each condition relative to the vehicle control of that strain.

For estradiol pre-conditioning (*Figure 6E,F* and *Figure 6—figure supplement 2*), overnight cultures grown in SCD were back diluted 1:100 in fresh SCD to ensure mock and estradiol cultures were at the same starting dilution. The culture was immediately split into two flasks (20 ml each), and one was treated with 20 µl 2 µM estradiol (final concentration 2 nM). Mock and estradiol-treated cultures were grown for 3 hr and then treated with DMSO (vehicle), 8 µg/ml DZA, or 2.5 mM AZC, grown for an additional 21 hr, and $OD_{600}$ measured. 'Relative growth' is the $OD_{600}$ for each condition relative to the mock (no estradiol), DMSO only control. Cultures were also assessed for relative cell size distribution by measuring side scatter on a Stratedigm S1000EX cytometer.

Serial dilution plating assay (*Figure 6A*) was performed by diluting overnight cultures to $OD_{600}$ ~ 1.0 in fresh media and serially diluting 1:10 on a 96-well plate. The cultures were stamped onto plates using a 'frogger' device and grown as indicated.

Thermotolerance (*Figure 2—figure supplement 2*) was performed by diluting overnight cultures to OD ~0.05 and growing for 5.5 hr. The culture was split and treated with the indicated concentrations of DZA for 45 min. One milliliter was removed and immediately placed on ice as a pre-heat shock control. One milliliter was placed at 50°C for 15 min on a heat block with thorough mixing every 5 min and then placed on ice. Cells were serially diluted 1:10,000 (for pre-heat shock cultures) or 1:100 (for post-heat shock cultures) and 200 µl were spread onto YPD plates. Plates were incubated at 30°C for 2 days and colonies were counted. Reported are the number of colonies formed on each post-heat shock plate, which corresponds to approximately 100,000 cells that were exposed to heat shock as determined from the pre-heat shock plates.

## Fluorescence microscopy

Preparing anchor-away strains expressing FRB-GFP–tagged proteins for microscopy was performed as described (*Haruki et al., 2008*). Briefly, 1 ml of cells was harvested, fixed in 1 ml −20°C methanol for 6 min, and resuspended in TBS/0.1% Tween with DAPI. Fixed, DAPI-stained cells were spotted onto a 2% agarose pad on a glass slide and topped with a cover slip. Samples were imaged for both

GFP and DAPI on a Nikon Ti2 microscope with a 100x objective and an ORCA-R2 cooled CCD camera (Hamamatsu).

Confocal microscopy of Sis1-YFP, Cfi1-mKate, and Hsp104-mKate was performed live by allowing low density cultures grown in SCD at room temperature to settle in 96-well glass bottom plates coated with concanavalin A. For treatments, medium was removed and fresh SCD containing the indicated drug was added to the well. Imaging was performed on a Nikon Ti microscope with a 100 × 1.49 NA objective, a spinning disk confocal setup (Andor Revolution) and an EMCCD camera (Andor).

## RNA-seq

RNA was depleted of ribosomal RNA using Yeast Ribo-Zero Gold (Illumina). For all auxin-related experiments, libraries were prepared from biological duplicates (individual strain isolates grown and treated on separate days) using the TruSeq Stranded Kit (Illumina). The diamide RNA-seq data are of libraries were prepared using another RNA-seq library construction protocol, as previously described (*Couvillion et al., 2016*) and were not done in replicate as the RNA-seq data recapitulated the well-characterized transcriptional response to diamide (*Gasch et al., 2000*). All libraries were sequenced on an Illumina NextSeq platform.

## RNA-seq data analysis

Raw fastq files were processed as follows. The adapter sequence (AGATCGGAAGAG) was removed using Cutadapt (v1.8.3) with option '-m 18' to retain reads >18 nt. Reads were then quality-filtered using PRINSEQ and alignment was performed with TopHat (v2.1.0). The resulting BAM files from each lane on the flow cell were merged, sorted, and indexed with SAMtools. The number of reads for each genomic feature (e.g. transcript), was quantified using HTSeq count. The GTF file was ENSEMBL release 91 for *Saccharomyces cerevisiae*.

Quantification and differential expression for auxin experiments were carried out using DESeq2 (*Love et al., 2014*) with drug treatment as the variable: two biological replicates each of mock-treated and auxin-treated. RNA abundance changes were reported using the $log_2$ fold change calculated by DESeq2 for auxin/untreated for each transcript. For ±diamide datasets, RNA abundance was determined using RPKM and reported as $log_2$ fold change (diamide vs. untreated) for each transcript. Quantified RNA-seq data can be found in *Supplementary file 4*.

Transcript classes were defined as follows. 'Hsf1 targets': identified using an approach that defines transcripts that fail to be activated when Hsf1 is depleted prior to acute heat shock (*Pincus et al., 2018*). 'Msn2/4 targets': classification from *Solís et al. (2016)*. 'All others': all other genes characterized as 'Verified ORFs' by SGD, excluding those in 'Hsf1 targets' and 'Msn2/4 targets' classes. 'Proteasome subunits': the 27 genes encoding the 27 subunits of the 26S proteasome. 'R-protein genes': the 136 genes encoding the 79 subunits of the ribosome (ribosomal proteins). 'Other ribosome biogenesis (RiBi) genes': 169 unique genes from the SGD GO term 'ribosome biogenesis' with r-protein genes removed. 'Hac1-dependent UPR genes': core set of UPR genes induced by Hac1 overexpression, tunicamycin treatment, and DTT treatment (*Pincus et al., 2014*). Gene lists can be found in *Supplementary file 4*.

## Data availability

All sequencing data has been deposited on Gene Expression Omnibus under accession number GSE114077.

## Acknowledgements

We thank members of the Churchman lab, F Winston, R Kingston, and M Sonnett for helpful discussions; S Doris, E McShane, and C Patil for critical reading of the manuscript; and T Powers, F Holstege, V Denic, and D Gross for reagents. Microscopy was performed at the Nikon Imaging Center at Harvard Medical School and the WM Keck Microscopy Facility at the Whitehead Institute. RNA-seq library preparation and sequencing was performed at the Whitehead Institute and Biopolymers Facility at Harvard Medical School, respectively. This work was supported by the NIH (R01-HG007173 and R01-GM117333 to LSC, R01-GM120122 to MS, R35-GM128813 to MW), the DOE (DE-SC0018420 to MW), and the NSF (Graduate Research Fellowship to BWT).

## Additional information

### Funding

| Funder | Grant reference number | Author |
|---|---|---|
| National Science Foundation | 2013171680 | Blake W Tye |
| National Institutes of Health | R35-GM128813 | Martin Wühr |
| Department of Energy, Labor and Economic Growth | DE-SC0018420 | Martin Wühr |
| National Institutes of Health | R01-GM120122 | Michael Springer |
| National Institutes of Health | R01-HG007173 | L Stirling Churchman |
| National Institutes of Health | R01-GM117333 | L Stirling Churchman |

The funders had no role in study design, data collection and interpretation, or the decision to submit the work for publication.

### Author contributions

Blake W Tye, Conceptualization, Funding acquisition, Investigation, Visualization, Writing—original draft, Writing—review and editing; Nicoletta Commins, Lillia V Ryazanova, Investigation; Martin Wühr, Supervision, Investigation; Michael Springer, Resources; David Pincus, Resources, Investigation; L Stirling Churchman, Conceptualization, Supervision, Funding acquisition, Writing—original draft, Writing—review and editing

### Author ORCIDs

Blake W Tye http://orcid.org/0000-0003-0841-6249
Martin Wühr http://orcid.org/0000-0002-0244-8947
David Pincus http://orcid.org/0000-0002-9651-6858
L Stirling Churchman http://orcid.org/0000-0003-3888-2574

### Decision letter and Author response

Decision letter https://doi.org/10.7554/eLife.43002.030
Author response https://doi.org/10.7554/eLife.43002.031

## Additional files

### Supplementary files

• Supplementary file 1. Yeast strains used in this study.
DOI: https://doi.org/10.7554/eLife.43002.021

• Supplementary file 2. Plasmids used in this study.
DOI: https://doi.org/10.7554/eLife.43002.022

• Supplementary file 3. Primers used in this study.
DOI: https://doi.org/10.7554/eLife.43002.023

• Supplementary file 4. Gene annotation lists and RNA-seq data used in *Figures 1–3*. Tab 'Gene_Lists' contains members of groups used for analysis. Subsequent tabs contain RNA abundance measurements determined by DESeq2 or RPKM calculations.
DOI: https://doi.org/10.7554/eLife.43002.024

• Supplementary file 5. Flow cytometry data from competitive fitness experiments used in *Figure 6*. Query (mCh) and reference (YFP) counts for each competition at t = 0, 1, 2, 3, 4, 5 days. Each mutant query had four isolates ('Iso1-4') that were tested in two technical replicates ('Rep1-2'), for a total of eight replicates per experiment. The normalized, $\log_{10}$ transformed values were used to generate plots.
DOI: https://doi.org/10.7554/eLife.43002.025

• Supplementary file 6. Summary of proteomics data of input and pellet proteins. The value of each protein is normalized to the total signal in each sample (TMT channel) to determine relative abundance within each sample (parts per million, ppm).
DOI: https://doi.org/10.7554/eLife.43002.026

• Transparent reporting form
DOI: https://doi.org/10.7554/eLife.43002.027

### Data availability

All sequencing data have been deposited on Gene Expression Omnibus under accession number GSE114077.

The following dataset was generated:

| Author(s) | Year | Dataset title | Dataset URL | Database and Identifier |
|---|---|---|---|---|
| Tye BW, Churchman LS | 2019 | Proteotoxicity from aberrant ribosome biogenesis compromises cell fitness | https://www.ncbi.nlm.nih.gov/geo/query/acc.cgi?acc=gse114077 | NCBI Gene Expression Omnibus, GSE114077 |

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
