## [Decision Letter]

Thank you for submitting your article "A risk-reward tradeoff of high ribosome production in proliferating cells" for consideration by *eLife*. Your article has been reviewed by three peer reviewers, including Alan Hinnebusch as the Reviewing Editor, and the evaluation has been overseen by Naama Barkai as the Senior Editor. The following individual involved in the review of your submission has agreed to reveal his identity: David Tollervey (Reviewer #2).

The reviewers have discussed the reviews with one another and the Reviewing Editor has drafted this decision to help you prepare a revised submission.

Summary:

The results in this study show that impairment of ribosome processing in budding yeast by various genetic and pharmacological interventions evokes induction of the Hsf1 regulon, without triggering the more general environmental stress response (ESR). Evidence is presented that it is the accumulation of newly synthesized unassembled (orphan) ribosomal proteins (RPs), most likely in aggregates, that is the triggering event, which evokes a reduction in RP gene (RPG) expression and reduced translation independently of the TOR or Gcn2 signalling pathways. Apart from this, they show that ribosomal protein assembly stress (dubbed RPAS) evokes altered nuclear localization of an Hsp70 co-chaperone (Sis1) and a nucleolar marker (Cfl1) and altered cytoplasmic localization of chaperone Hsp104, from which they conclude that both nuclear and cytosolic proteostasis is disrupted. Consistent with the last conclusion, both Hsf1 and the transcription factor Rpn4, which induces proteasome formation, are required for growth per se (Hsf1) or cellular fitness in a growth competition assay (Rpn4) under conditions of RPAS. These findings are significant in revealing previously unknown cellular consequences of impairing ribosome biogenesis in yeast, including specific induction of the Hsf1 regulon (and not the stereotypical ESR), and in down-regulating translation independently of TOR/Gcn2. These findings are relevant to the cellular basis of ribosomopothies, and other stresses or disease states that disrupt ribosome biogenesis in mammals.

Essential revisions:

1) It was noted that previous results were published indicating that yeast and human cells degrade ribosomal proteins produced in excess, and that cells lacking this quality control mechanism contain aggregated ribosomal proteins (McShane et al., 2016; Sung et al., 2016a, 2016b). Hence, the most novel findings of the current paper involve the specific induction of the Hsf1 regulon (and not the stereotypical ESR), and the down-regulation of ribosomal protein mRNAs and bulk translation independently of TOR/Gcn2. As such, the reviewers feel that the results of these previous publications should be cited in the Introduction, and that additional experiments should be carried out to further elucidate the novel stress responses uncovered here, as follows:

- New experiments are likely required to investigate whether the specific induction of Hsf1, without eliciting the general environmental stress response, is triggered by accumulation of pre-ribosomes in the nucleolus/nucleus or only by the accumulation/aggregation of excess ribosomal proteins in the cytoplasm.

- It is also important to investigate whether the reduced expression of RPG mRNAs observed here is a common response to proteotoxic stress, e.g. induced by AZC, or rather is a more specific response to accumulation/aggregation of excess ribosomal proteins.

2) It was felt that because no evidence was provided for ribosomal over-production or aggregation in wild-type cells under conditions of high versus low ribosome production, the title of the paper, statement in the Abstract that ribosome assembly is a "lynchpin of cellular homeostasis", and certain related statements in the Discussion appear to be overinterpretations. Either they should provide such evidence or revise the paper to correct these overstatements. Also, the Abstract should be modified to eliminate the claim that overproduction of rRNAs has been examined in this study.

All other comments raised by the reviewers should be addressed with suitable alterations of text, or an explanation as to why no alterations are required.

*Reviewer #1:*

This is a very well-designed, well-executed and interesting study showing that impairment of ribosome processing in budding yeast by various genetic and pharmacological interventions evokes induction of the Hsf1 regulon, without triggering the more general environmental stress response (ESR). Evidence is presented that it is the accumulation of newly synthesized unassembled (orphan) ribosomal proteins (RPs), most likely in aggregates, that is the triggering event, which evokes a reduction in RP gene (RPG) expression and reduced translation independently of the TOR or Gcn2 signalling pathways. (The mechanism for the latter response is left unresolved.) Apart from this, they show that ribosomal protein assembly stress (dubbed RPAS) evokes altered nuclear localization of an Hsp70 co-chaperone (Sis1) and a nucleolar marker (Cfl1) and altered cytoplasmic localization of chaperone Hsp104, from which they conclude that both nuclear and cytosolic proteostasis is disrupted. Consistent with the last conclusion, both Hsf1 and the transcription factor Rpn4, which induces proteasome formation, are required for growth per se (Hsf1) or cellular fitness in a growth competition assay (Rpn4) under conditions of RPAS.

These findings are significant in revealing previously unknown cellular consequences of impairing ribosome biogenesis in yeast, including specific induction of the Hsf1 regulon (and not the stereotypical ESR), and in down-regulating translation independently of TOR/Gcn2. These findings are relevant to the cellular basis of ribosomopothies, and other stresses or disease states that disrupt ribosome biogenesis in mammals. The experiments are generally incisive and the quality of the data is uniformly high. However, there are a few issues to be addressed concerning the evidence for a general induction of nuclear and cytoplasmic loss of proteostasis, which seems inadequate. Also, the notion that the process of ribosome biosynthesis is inherently susceptible to conferring loss of proteostasis and puts cells at risk at high growth rates is an interesting idea, but not really well enough established by data to serve as the title of the paper.

- In Figure 4D, owing to the reduced general translation in DZA-treated cells, only Rps26 shows accumulation in aggregates at levels higher than in the DMSO-treated control cells, meaning that only 1 of the five RPs tested in Figure 4B-F can be said to accumulate in aggregates in cells with impaired ribosome biogenesis. One wonders if this is sufficient to disrupt nuclear and cytosolic proteostasis. The latter is being inferred primarily by the relocalization of Sis1-YFP and Cfi1-mFate within the nucleus and Hsp104-Kate in the cytoplasm, but are these phenotypes incisive indicators of the disruption of nuclear and cytosolic proteostasis? Wouldn't assaying the activity of a protein that is prone to sequestration in aggregates of unfolded proteins greatly complement these cell biological phenotypes? In addition, one would like to see that these cellular phenotypes are also triggered by inducing RPAS genetically rather than pharmacologically, e.g. with the rat1-AID strain, and to know whether they are triggered by any other known inducers of proteotoxic stress, e.g. AZC, and if not why? It also seems important to determine whether the unfolded protein response is triggered by RPAS, which can be assayed with a reporter gene, which could potentially provide additional evidence for loss of cytoplasmic proteostasis.

- Discussion, first paragraph: They conclude that induction of proteotoxic stress by RPAS compromises other cellular processes, but what are they? Is the reduced expression of RPG mRNAs and reduced translation a common response to proteotoxic stress, e.g. is it induced by AZC, or is this a more specific regulatory response to decreased ribosome biogenesis capacity that is unrelated to proteotoxic stress? As indicated above, one would like to see evidence that some other cellular process commonly impaired by loss of proteostasis is compromised by DZA treatment to the same/similar degree as observed in response to AZC or heat-shock.

- The title of the paper and text in the first paragraph of the Discussion, describes the interesting notion that the high demands of proliferating cells for ribosomes puts them at risk of disrupting proteostasis. However, it's not clear that the data support this notion. Relatively low levels of certain unassembled RPs appears to be sufficient to induce Hsf1 (Figure 4D), and this might well occur at low levels of ribosome assembly on a relatively poor carbon or nitrogen source compared to glucose medium with plenty of nitrogen. Can they show that lower levels of DZA are sufficient to induce Hsf1 in cells growing in rich medium vs. poor medium where the rates of ribosome production markedly differ? If not, they should consider choosing a different, less provocative title for the paper that is captures the findings that are best established by the data.

*Reviewer #2:*

The results are interesting and I think that the conclusions are likely to be generally correct. The question of induction of Hsf1 etc. by pre-ribosome accumulation should be addressed. However, I expect that the authors will be able to satisfactorily argue this point.

Not sure about the title, which sounds striking but actually seems a bit tangential to the topic of the paper.

The authors report that accumulation of ribosomal proteins in excess of the requirements for ribosome production is toxic through aggregation. The conclusions are of interest and the manuscript is suitable for *eLife* if suitable revised.

1) Figure 5: It might have been anticipated that RPs synthesized in excess of their escortins, would accumulate in the cytoplasm. DZA treatment blocks a late step in 60S subunit synthesis and it seems possible that the altered nuclear localization of co-chaparones reflects association with defective pre-ribosomes. This raises the question of whether the defects induced by ribosome synthesis inhibition primarily reflect RP or pre-ribosome accumulation. This concern should be addressed more explicitly in the revised manuscript.

*Reviewer #3:*

Tye et al. studied the effects of perturbations of ribosome protein and RNA levels on protein homeostasis. They determined that disruption of the formation of properly assembled ribosomes leads to activation of the heat shock response (HSR). This was determined to be likely due to aggregation of ribosomal proteins (r-proteins). Additionally, the effect of r-protein aggregation on cellular fitness was exacerbated by proteasome impairment.

The paper does a good job of demonstrating that having excess r-proteins unable to form proper ribosomes will aggregate and induce the HSR. Several major conclusions of the paper, however, are not supported in the data. The title states that there is a "risk-reward tradeoff of high ribosome production," yet provides no evidence that high ribosome production leads to an excess of r-proteins or the associated stress described in the paper. The Abstract states that "imbalances in synthesis" between the r-proteins and rRNAs leads to a compromise of proteostasis. However, it is explicitly an overproduction of r-proteins (and not rRNAs) that leads to HSR activation. This is less novel, as it has been demonstrated (as noted in the paper for mitochondrial complexes) that imbalances in protein complex stoichiometry will often lead to misfolded protein and therefore a stress response. The only conditions in which the described stress is seen is in conditions of artificially limiting rRNA levels or artificially disrupting complex formation. Given that there is no evidence of r-protein overproduction occurring during high ribosomal production (as stated in the title) or in any stress condition beyond specific disruption of ribosomes, it is an overreach to call ribosome assembly a "lynchpin of cellular homeostasis" (as stated in the Abstract).

The central premise that cells must balance the reward of high ribosome production with the risk of aggregation of excess r-proteins is intriguing, however it is critical to show that this stress exists in conditions beyond the specific case of directly manipulating ribosome stoichiometry or assembly. Specifically, the authors must show that this stress occurs under conditions of high ribosome production (or too-high ribosome production) and is alleviated when ribosome production drops. This would show the tradeoff between high ribosome production and proteotoxic stress implied by the title. In current form, the scope of the data is limited to a thorough and convincing demonstration that excess r-proteins will aggregate and activate Hsf1 if not properly assembled into ribosomes.

---

## [Author Response]

Essential revisions:1) It was noted that previous results were published indicating that yeast and human cells degrade ribosomal proteins produced in excess, and that cells lacking this quality control mechanism contain aggregated ribosomal proteins (McShane et al., 2016; Sung et al., 2016a, 2016b). Hence, the most novel findings of the current paper involve the specific induction of the Hsf1 regulon (and not the stereotypical ESR), and the down-regulation of ribosomal protein mRNAs and bulk translation independently of TOR/Gcn2. As such, the reviewers feel that the results of these previous publications should be cited in the Introduction, and that additional experiments should be carried out to further elucidate the novel stress responses uncovered here, as follows:

We agree with the recommendation and have moved these references to the Introduction.

- New experiments are likely required to investigate whether the specific induction of Hsf1, without eliciting the general environmental stress response, is triggered by accumulation of pre-ribosomes in the nucleolus/nucleus or only by the accumulation/aggregation of excess ribosomal proteins in the cytoplasm.

We have performed new experiments to broadly investigate the proteins that aggregate in response to ribosomal protein assembly stress. We analyzed aggregates from DZA-treated cells by quantitative mass spectroscopy and found enrichment of a broad complement of r-proteins, including those that assemble in the nucleolus (see Figure 4G, H of revised manuscript). These data suggest that following disruption of ribosome assembly, orphan r-proteins aggregate both in the nucleus and cytosol and serve as the most likely Hsf1 activation signal.

We do not believe that pre-ribosomes directly elicit the Hsf1 response for several reasons:

1) Many lines of evidence point towards Hsf1 activation being predicated on accumulation of misfolded/aggregated proteins that titrate chaperones such as Hsp70 away from binding and inactivating Hsf1 (Shi et al., Genes Dev, 1998; Zheng et al., *eLife* 2016). Thus we cannot envision how accumulation of precursor ribosome complexes *per se* would lead to Hsf1 activation.

2) The RPAS response occurs from depletion of rRNA processing factors, which therefore depletes the platform (rRNA) for precursor assembly. In this case, ribosomal proteins cannot assemble and will thus dominate the triggering signal.

3) Depletion of ribosomal protein escortins elicits the response, presumably directly from the escortin clients (single r-proteins).

Nevertheless, as we cannot directly test whether a pre-ribosome could contribute to activating Hsf1, we now mention the possibility in the text (Discussion, first paragraph).

- It is also important to investigate whether the reduced expression of RPG mRNAs observed here is a common response to proteotoxic stress, e.g. induced by AZC, or rather is a more specific response to accumulation/aggregation of excess ribosomal proteins.

RPG expression is commonly repressed across the vast majority of stress conditions that we are aware of, including proteotoxic stressors such as oxidative stress, heat (Gasch et al., 2000) and AZC (Trotter et al., 2002). Indeed, in our oxidative stress control (diamide) we found repression of RPGs. We think this is an important point and we raise it in the text (subsection “Compromised r-protein gene expression and translational output during RPAS”). In particular, we describe how most stress responses repress RPGs *and* ribosome biogenesis genes. By contrast, RPAS only represses RPGs.

2) It was felt that because no evidence was provided for ribosomal over-production or aggregation in wild-type cells under conditions of high versus low ribosome production, the title of the paper, statement in the Abstract that ribosome assembly is a "lynchpin of cellular homeostasis", and certain related statements in the Discussion appear to be overinterpretations. Either they should provide such evidence or revise the paper to correct these overstatements. Also, the Abstract should be modified to eliminate the claim that overproduction of rRNAs has been examined in this study.

Supporting our model, we have added new experimental results indicating that cells making fewer ribosomes experience a milder proteostatic stress during RPAS (Figure 7A, B). Specifically, we studied environmental (suboptimal carbon source, glycerol, as suggested by reviewer 1) and genetic (*sch9*Δ) strategies to slow the production of ribosomes. In both cases, we find a reduced activation levels of Hsf1 target genes when treated with DZA.

These data demonstrate a relationship between ribosome production levels and the risk of proteostasis collapse upon disruption of ribosome biogenesis. Nevertheless, we have decided to change the Title/Abstract to ensure our findings are clearly communicated.

Reviewer #1:

[…]There are a few issues to be addressed concerning the evidence for a general induction of nuclear and cytoplasmic loss of proteostasis, which seems inadequate. Also, the notion that the process of ribosome biosynthesis is inherently susceptible to conferring loss of proteostasis and puts cells at risk at high growth rates is an interesting idea, but not really well enough established by data to serve as the title of the paper.- In Figure 4D, owing to the reduced general translation in DZA-treated cells, only Rps26 shows accumulation in aggregates at levels higher than in the DMSO-treated control cells, meaning that only 1 of the five RPs tested in Figure 4B-F can be said to accumulate in aggregates in cells with impaired ribosome biogenesis. One wonders if this is sufficient to disrupt nuclear and cytosolic proteostasis.

The reviewer highlights the difficulty of quantitatively analyzing the western blotting data. Due to a varying background of pelleting proteins, it is necessary to compare the pellet to the input band intensities for each experiment as we do in Figure 4. Comparing the band intensities of the pellet across samples is not informative.

To supplement these data, we now include new experiments to obtain a broader understanding of how RPAS disrupts proteostasis. As described above in “Essential Revisions”, we applied quantitative proteomics to analyze aggregates from DZA-treated cells to obtain an unbiased view of which proteins aggregate in their wild-type form. These experiments reveal that r-proteins as a whole are enriched in the pellet after DZA treatment (see Figure 4G).

The latter is being inferred primarily by the relocalization of Sis1-YFP and Cfi1-mFate within the nucleus and Hsp104-Kate in the cytoplasm, but are these phenotypes incisive indicators of the disruption of nuclear and cytosolic proteostasis?

Yes. Hsp104 functions in protein disaggregation at cytosolic quality control (Q-) bodies and insoluble protein deposits (IPOD), and is widely used as a marker for all types of protein aggregation under many conditions of proteotoxic stress (Glover and Lindquist, 1998; Tkach and Glover, 2004; Kaganovich, Kopito and Frydman, 2008; Zhou et al., Cell, 2014). The Hsp70 co-chaperone Sis1 functions in clearing misfolded proteins that accumulate in the nucleus at subnuclear bodies termed the intranuclear quality control compartment (INQ) (Summers et al., 2013; Park et al., 2013; Hill, Hanzen and Nystrom, 2017). Furthermore, relocalization of Sis1 to subnuclear sites under proteotoxic stressors such as heat shock has been previously reported (Malinovska et al., MBoC, 2012).

Wouldn't assaying the activity of a protein that is prone to sequestration in aggregates of unfolded proteins greatly complement these cell biological phenotypes?

The cell biological phenotypes we observe by microscopy are now corroborated by new data. Analysis of RPAS-induced aggregates by mass spectrometry detects a number of nucleolar ribosome biogenesis factors, including Nop53, Rrp8, Mak16, and others (see Figure 4H of the revised manuscript). We also see a 5-fold accumulation of the cytosolic co-chaperone Hsp42 in aggregates. These data and the microscopy results strongly indicate disrupted proteostasis in the nucleus and cytosol.

In addition, one would like to see that these cellular phenotypes are also triggered by inducing RPAS genetically rather than pharmacologically, e.g. with the rat1-AID strain, and to know whether they are triggered by any other known inducers of proteotoxic stress, e.g. AZC, and if not why?

We feel that a detailed investigation of the cellular phenotypes under many conditions, while interesting, represents an expansive topic that we hope the reviewer understands is outside the scope of this manuscript. Furthermore, analysis of some of these cellular phenotypes after AZC exposure has already been performed and reported (Zhou et al., Cell, 2014).

Technical strain construction challenges have not allowed assaying the AID strains by microscopy. However, it is our view that DZA treatment canonically induces RPAS; we have demonstrated that DZA treatment phenocopies inducible degradation perturbations very well in all assays where it can be compared to other RPAS-inducing conditions.

It also seems important to determine whether the unfolded protein response is triggered by RPAS, which can be assayed with a reporter gene, which could potentially provide additional evidence for loss of cytoplasmic proteostasis.

We appreciate the suggestion. We do not observe activation of the canonical endoplasmic reticulum unfolded protein response (UPR) (analysis now presented in Figure 2—figure supplement 3), underscoring the specificity of the response.

- Discussion, first paragraph: They conclude that induction of proteotoxic stress by RPAS compromises other cellular processes, but what are they? Is the reduced expression of RPG mRNAs and reduced translation a common response to proteotoxic stress, e.g. is it induced by AZC, or is this a more specific regulatory response to decreased ribosome biogenesis capacity that is unrelated to proteotoxic stress?

This is an important point that we have now clarified in the text. Briefly, as discussed above in “Essential Revisions”, RPG expression is commonly repressed across the vast majority of stress conditions, including proteotoxic stressors such as oxidative stress, heat (Gasch et al., 2000) and AZC (Trotter et al.,2002). Indeed, in our oxidative stress control (diamide) we found repression of RPGs. We raise this point this in the text (subsection “Compromised r-protein gene expression and translational output during RPAS”) where we describe how most stress responses repress RPGs *and* ribosome biogenesis genes. By contrast, RPAS only represses RPGs.

Reduced translation is also observed in a number of stress responses, such as glucose starvation, amino acid starvation, DNA damage, and peroxide, which is controlled by canonical signaling pathways (Cherkasova and Hinnebusch, 2003; Shenton et al., 2006). By contrast, we found that RPAS compromises r-protein gene expression and translation independent of these signaling pathways.

As indicated above, one would like to see evidence that some other cellular process commonly impaired by loss of proteostasis is compromised by DZA treatment to the same/similar degree as observed in response to AZC or heat-shock.

We are unclear on what cellular process the reviewer would like to see compromised. Our data demonstrate that protein homeostasis is disrupted at levels that mirror perturbations such as heat, oxidative stress, or AZC; Hsf1 activation level and markers like Hsp104 foci formation are comparable for RPAS and other stresses. We have also shown that bolstering the protein homeostasis network (through exogenously inducing Hsf1 targets) partly rescues the RPAS growth phenotype, indicating that the loss of protein homeostasis directly contributes to the reduction in cell growth.

- The title of the paper and text in the first paragraph of the Discussion, describes the interesting notion that the high demands of proliferating cells for ribosomes puts them at risk of disrupting proteostasis. However, it's not clear that the data support this notion. Relatively low levels of certain unassembled RPs appears to be sufficient to induce Hsf1 (Figure 4D), and this might well occur at low levels of ribosome assembly on a relatively poor carbon or nitrogen source compared to glucose medium with plenty of nitrogen. Can they show that lower levels of DZA are sufficient to induce Hsf1 in cells growing in rich medium vs. poor medium where the rates of ribosome production markedly differ? If not, they should consider choosing a different, less provocative title for the paper that is captures the findings that are best established by the data.

We appreciate the suggestion to distinguish between scenarios of high versus low ribosome production. As described above in “Essential revisions”, we have added new experimental results indicating that cells making fewer ribosomes experience a milder proteostatic stress during RPAS (Figure 7A, B).

Reviewer #2:

The results are interesting and I think that the conclusions are likely to be generally correct. The question of induction of Hsf1 etc. by pre-ribosome accumulation should be addressed. However, I expect that the authors will be able to satisfactorily argue this point.Not sure about the title, which sounds striking but actually seems a bit tangential to the topic of the paper.The authors report that accumulation of ribosomal proteins in excess of the requirements for ribosome production is toxic through aggregation. The conclusions are of interest and the manuscript is suitable for eLife if suitable revised.1) Figure 5: It might have been anticipated that RPs synthesized in excess of their escortins, would accumulate in the cytoplasm. DZA treatment blocks a late step in 60S subunit synthesis and it seems possible that the altered nuclear localization of co-chaparones reflects association with defective pre-ribosomes. This raises the question of whether the defects induced by ribosome synthesis inhibition primarily reflect RP or pre-ribosome accumulation. This concern should be addressed more explicitly in the revised manuscript.

We discuss this point above in the “Essential revisions”, and copy it here. We do not believe that pre-ribosomes directly elicit the Hsf1 response for several reasons:

1) Many lines of evidence point towards Hsf1 activation being predicated on accumulation of misfolded/aggregated proteins that titrate chaperones such as Hsp70 away from binding and inactivate Hsf1due to the exposed hydrophobic sequences that recruit chaperones (Shi et al., Genes Dev, 1998; Zheng et al., *eLife* 2016). Thus we cannot envision how accumulation of precursor ribosome complexes *per se* would lead to Hsf1 activation.

2) The RPAS response occurs from depletion of rRNA processing factors, which therefore depletes the platform (rRNA) for precursor assembly.

3) Depletion of ribosomal protein escortins elicits the response, presumably from the escortin clients (single r-proteins) directly.

Nevertheless, as we cannot directly test whether a pre-ribosome could contribute to activating Hsf1, we now mention the possibility in the text (Discussion, first paragraph).

Reviewer #3:

[…] The paper does a good job of demonstrating that having excess r-proteins unable to form proper ribosomes will aggregate and induce the HSR. Several major conclusions of the paper, however, are not supported in the data. The title states that there is a "risk-reward tradeoff of high ribosome production," yet provides no evidence that high ribosome production leads to an excess of r-proteins or the associated stress described in the paper. The Abstract states that "imbalances in synthesis" between the r-proteins and rRNAs leads to a compromise of proteostasis. However, it is explicitly an overproduction of r-proteins (and not rRNAs) that leads to HSR activation. This is less novel, as it has been demonstrated (as noted in the paper for mitochondrial complexes) that imbalances in protein complex stoichiometry will often lead to misfolded protein and therefore a stress response. The only conditions in which the described stress is seen is in conditions of artificially limiting rRNA levels or artificially disrupting complex formation. Given that there is no evidence of r-protein overproduction occurring during high ribosomal production (as stated in the title) or in any stress condition beyond specific disruption of ribosomes, it is an overreach to call ribosome assembly a "lynchpin of cellular homeostasis" (as stated in the Abstract).

Disruptions to protein complex assembly has been shown to activate proteostatic restoration mechanisms, not the entire Hsf1 regulon. The work on mitochondrial protein complexes cited by the reviewer describes an activation of the proteasome as a cellular coping strategy, but does not describe any action of the Hsf1 regulon. We believe our work highlights ribosome assembly as a particular vulnerability to cytosolic/nuclear proteostasis and a feasible part of the driving force for Hsf1 activity.

In any case, we have altered the Title and Abstract as suggested by the reviewers, to ensure our findings are clearly communicated.

The central premise that cells must balance the reward of high ribosome production with the risk of aggregation of excess r-proteins is intriguing, however it is critical to show that this stress exists in conditions beyond the specific case of directly manipulating ribosome stoichiometry or assembly. Specifically, the authors must show that this stress occurs under conditions of high ribosome production (or too-high ribosome production) and is alleviated when ribosome production drops. This would show the tradeoff between high ribosome production and proteotoxic stress implied by the title. In current form, the scope of the data is limited to a thorough and convincing demonstration that excess r-proteins will aggregate and activate Hsf1 if not properly assembled into ribosomes.

We thank the reviewer for raising these points about the risk-reward relationship. As described above under “Essential revisions”, we have added new experimental results indicating that cells making fewer ribosomes experience a milder proteostatic stress during ribosome protein assembly stress (Figure 7A, B).